

**Future permafrost degradation under climate change in a headwater catchment of Central**
**Siberia: quantitative assessment with a mechanistic modelling approach**
Thibault Xavier[1], Laurent Orgogozo[1*], Anatoly S. Prokushkin[2], Esteban Alonso-González[3], Simon
Gascoin[4], Oleg S. Pokrovsky[1,5]
[1]Geoscience Environnement Toulouse (GET), CNRS, UMR5563, Toulouse, 31400, France
[2]V.N. Sukachev Institute of Forest SB RAS, Russia
[3]Instituto Pirenaico de Ecología, Consejo Superior de Investigaciones Científicas (IPE-CSIC), Jaca,
Spain
[4]Centre d'Etudes Spatiales de la Biosphère, Université de Toulouse, CNRS/CNES/IRD/INRA/UPS,
Toulouse, France
[5]BIO-GEO-CLIM Laboratory, Tomsk State University, Tomsk, Russia
*Corresponding author*: Laurent Orgogozo (laurent.orgogozo@get.omp.eu)
**Abstract**
Permafrost thawing as a result of climate change has major consequences locally and globally for
the biosphere as well as for human activities. The quantification of its extent and dynamics under
different climate scenarios is needed to design local adaptation and mitigation measures and to
better understand permafrost climate feedbacks. To this end, numerical simulation can be used to
explore the response of soil thermo-hydric regimes to changes in climatic conditions. Mechanistic
approaches minimize modelling assumptions by relying on the numerical resolution of continuum
mechanics equations, but involve significant computational effort. In this work, the permaFoam
solver is used along with high-performance computing resources to assess the impact of four
climate scenarios of the Coupled Model Intercomparison Project - Phase 6 (CMIP6) on permafrost
dynamics within a pristine, forest-dominated watershed in the continuous permafrost zone. Using
these century time-scale simulations, changes in soil temperature, soil moisture, active layer
thickness and water fluxes are quantified, assuming no change in vegetation cover. The most severe
scenario (SSP5-8.5) suggests a dramatic increase in both active layer thickness and annual
evapotranspiration, with maximum values on the watershed reached in 2100 of +46% and +29%
respectively. For the active layer thickness, in current climatic conditions it would correspond to a
560 km southward shift. Moreover, in this scenario thermal equilibrium of near-surface permafrost
with the new climatic conditions would not be reached in 2100, suggesting a further thawing of
permafrost even in case of halting the climate change.





**Keywords**
Permafrost, climate change, boreal forest, numerical modelling, high performance computing, soil
temperature, soil moisture, evapotranspiration.



## 1 Introduction

Permafrost is mostly situated in regions that are experiencing especially intense climate change, resulting in widespread warming and thawing with shrinking of its lateral extension and thickening of soil active layer (Biskaborn et al., 2019, Hu et al., 2022, Li et al., 2022a, 2022b). Permafrost thawing induces sizable changes in the environments (Walvoord and Kurylyk, 2016, Nitze et al., 2018, Makarieva et al., 2019, Jin et al., 2022, Wright et al., 2022) and for human activities (Shiklomanov et al., 2017, Strelestkiy et al., 2019, 2023, Hjort et al., 2018, 2022) in the Arctics and the sub-Arctics. For instance, permafrost-thaw related decrease of soil moisture leads to an increase in boreal fire frequency (Kurylyk, 2019, Kim et al., 2020), while soil mechanical instabilities induced by permafrost thawing threaten population settlements (Ramage et al., 2021) and infrastructures (Bartsch et al., 2021). Moreover, permafrost thaw may exert significant controls on biogeochemical cycles of carbon and related metals (Sonke et al., 2018, Karlsson et al., 2021, Walvoord and Striegl, 2021) and climate dynamics (Miner et al., 2022, Park and Kug, 2022, de Vrese et al., 2023), with potentially major feedback on climate warming. Thus anticipating the evolution of permafrost cover and dynamics is of primary importance for understanding and mitigating the climate change induced impacts at high latitudes. For this, robust and accurate numerical simulations are required (Schneider von Deimling et al., 2022, Hu et al., 2023b).

Boreal forest is one of the largest biome on Earth (Gauthier et al., 2015), and 80% of its area is located in permafrost regions (Stuenzi et al., 2021). Given that most permafrost areas are covered by boreal forest. Due to the complexity of the biophysical processes involved, quantifying permafrost dynamics evolution in boreal forests under climate change requires mechanistic, high-resolution modelling approaches (Orgogozo et al., 2019). Meanwhile, the large extent of the considered areas makes the use of such approaches impracticable at global, continental or regional scale. As a consequence, mechanistic modelling of permafrost dynamics has to focus on processes at the watershed scale in headwater catchments with long term environmental monitoring, following a general trend in Arctic sciences (Speetjens et al., 2023, Vonk et al., 2023). In Arctic environments, the vegetation strongly controls surface energy budget (Fedorov et al., 2019, Oehri et al., 2022), interacts with climate dynamics (Park et al., 2020, Kyrdyanov et al., 2024) and drives water fluxes (Orgogozo et al., 2019). As such, vegetation should be taken into account when simulating the impact of climate warming on permafrost in boreal forest areas (Loranty et al., 2018, Kirdyanov et al., 2020, Holloway et al., 2020).



Quantitative mechanistic modeling of permafrost dynamics under climate change at the headwater
catchment scale requires large computational resources, because fine spatio-temporal discretisations
are needed due to the strong non-linearities and couplings of the involved physics (Kurylyk and
Watanabe, 2013). This is especially important for century long simulation periods (O'Neill et al.,
2016) and simulation domains with surfaces up to tens of square kilometers (e.g.: Arndal and Torp-
Jørgensen, 2020). Therefore, high performance computing techniques are needed (Orgogozo et al.,
72  2023).

In this study, we focus on a permafrost-dominated, forested watershed of central Siberia
which was subjected to long term environmental monitoring, the Kulingdakan watershed (e.g.:
Prokushkin et al., 2007, Mashukov et al., 2021). The objective is to assess the future state of
permafrost and ground thermal regime in this continuous permafrost, boreal forest environment
under different climate change scenarios at the century time scale. The permafrost status of this
catchment under current climatic conditions has already been investigated (Orgogozo et al., 2019).
Here, we simulate, using a mechanistic modelling approach, the permafrost dynamics at the
catchment scale until 2100 under various scenarios of climate change. The vegetation controls on
permafrost dynamics are partly included in the mechanistic modelling framework, considering
evapotranspiration fluxes (Orgogozo et al., 2019), and partly handled empirically, via the
accounting on the insulating effect of ground floor vegetation (Blok et al., 2011, Cazaurang et al.,
2023). However, because no changes of vegetation is explicitly considered, we assume constant
biomass and primary production and therefore investigate only the physical part of the response of
permafrost to climate change. We use the permaFoam High Performance Computing
cryohydrogeological simulator (Orgogozo et al., 2023) with a national level supercomputing
infrastructure, the Joliot-Curie supercomputer of the Très Grand Centre de Calcul (TGCC) of the
French Alternative Energies and Atomic Energy Commission (CEA). The simulated permafrost
thawing features in Kulingdakan are discussed and compared for the different CMIP6 scenarios,
and the state and evolution of the thermal imbalance of the permafrost (e.g.: Ji et al., 2022, Nitzbon
et al., 2023) in the considered region.
**2 Materials and methods**
*2.1 Study site: Kulingdakan, a forested catchment in continuous permafrost area*
The Kulingdakan catchment is located in the Krasnoïarsk Region (64.31°N, 100.28°E),
within a continuous permafrost zone, belonging to the boreal forest biome (Northern Taïga – see



Figure 1a). This pristine catchment is monitored for the study of boreal processes over past two
decades. The vegetation is dominated by larch (*Larix gmelinii*), dwarf shrubs, mosses and lichens.
The catchment covers an area of 41 km² and has an elevation ranging from 132 m to 630 m
(Prokushkin et al., 2004). The climate is cold and continental, with an average annual mean
temperature of -8°C and annual total precipitation of 400 mm (annual mean measured between
1999 and 2014 at the Tura meteorological station, 5 km south of the Kulingdakan catchment, 168 m
altitude). The stream, which flows from east to west, divides the 41 km² catchment area into two
approximately rectangular slopes of equal area, the North Aspect Slope (NAS) and the South
Aspect Slope (SAS). The hydrological budget in this watershed is largely dominated by
evapotranspiration fluxes (Orgogozo et al., 2019). Two horizons constitute the soil in the first few
meters: an organic horizon (duff) and a mineral horizon (mainly rocky/gravely loam).
Due to the difference in solar radiation induced by their aspect, primary production and
evapotranspiration are more intensive in SAS than in NAS. Thus the two slopes show significant
differences, in larch trees size and larch stands density, as well as in rooting depth, organic horizon
thickness and moss layer thickness. The thickness of the organic horizon is of 11.6 cm on the NAS
and 7.7 cm on the SAS (Gentsch, 2011), while the moss layer thickness is of 13 cm on the NAS and
6.4 cm on the SAS (Prokushkin et al., 2007). The rooting depth is of 10 cm into the mineral horizon
for NAS, and 60 cm for SAS (Viers et al., 2013), and this difference has been shown to be of great
importance for the dynamics of the active layer (Orgogozo et al., 2019). These pedological and
physiological contrasts between the two aspects of the watershed slope, summarized in Figure 1b,
must be considered when performing permafrost simulations (Supplementary material B).



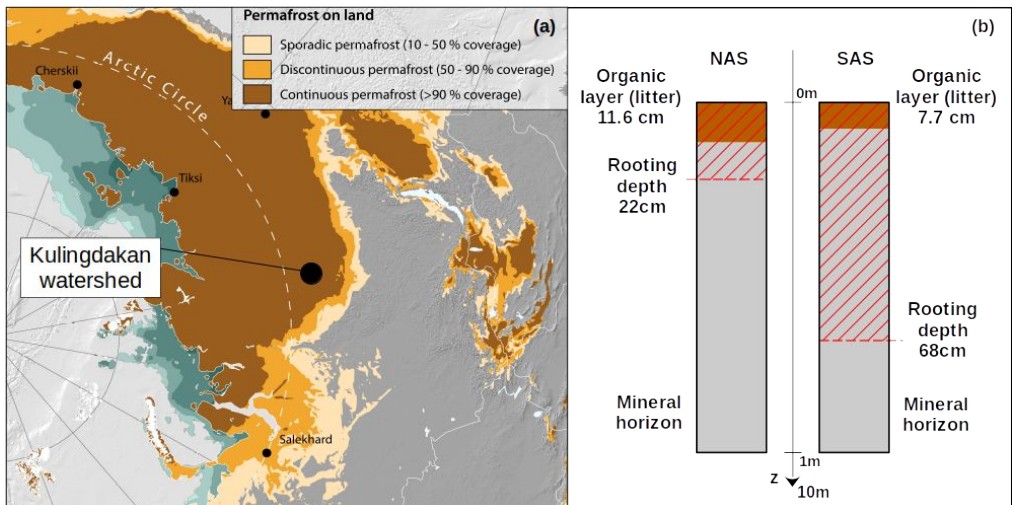

**Figure 1: (a) Location of Kulingdakan watershed (map from GRID-Arendal/Nunataryuk). (b) Representation of soil column structure for North aspected slope (NAS) and South aspected slope (SAS) of the Kulingdakan watershed.**

Previous modelling studies in the Kulingdakan catchment on water fluxes repartition, soil temperature at different depths and active layer thickness (Orgogozo et al., 2019, Orgogozo et al., 2023) demonstrated that the use of the permaFoam solver, together with boundary conditions (water fluxes and soil surface temperature) provided by field measurements, enabled to obtain numerical simulation results in agreement with in-situ observations under current climatic conditions .

### 2.2 The permaFoam cryohydrogeological simulator

The numerical tool used in this study is permaFoam (Orgogozo et al., 2019, 2023), the permafrost modelling solver developed in the framework of OpenFOAM, the open source, high performance computing tool box for computational fluid dynamics (Weller et al., 1998, openfoam.org, openfoam.com). This solver is designed to simulate 3D, transient coupled heat and water transfers in a variably saturated soil with evapotranspiration and freeze/thaw of the pore water. The two main equations solved by permaFoam are the Richards equation (1) that governs the flow of water and an energy balance equation (2) that governs the heat transfer, both defined at the Darcy scale of the considered porous medium (soil).



$$C_H(h)\frac{\partial h}{\partial t} = \nabla . \left( K_H(h,T) . \nabla(h+z) \right) + Q_{AET}(h,t) \tag{1}$$

$$\frac{\partial\left(\left(C_{T,eq}(h,T) + L\frac{\partial \theta_{ice}(h,T)}{\partial T}\right)T\right)}{\partial t} + \nabla . \left( V(h,T) C_{T,liquid} T \right) = \nabla . \left( K_{T,eq}(h,T) \nabla T \right) \tag{2}$$


The two primary variables in equations (1) and (2) are the generalized water pressure head $h$ [m]
and the soil temperature $T$ [K], respectively. In Richards equation (1), z is the vertical coordinate
[m] (oriented upward), $K_H$ is the hydraulic conductivity of the variably saturated, variably frozen
porous medium [m.s$^{-1}$], $C_H$ is the capillary capacity (also called specific moisture capacity) of the
unsaturated porous medium [m$^{-1}$] and $Q_{AET}$ [s$^{-1}$] is a source term representing the water uptake by
the vegetation through the evapotranspiration process (computed using Hamon formula, see Hamon
1963, Frolking, 1997). From the pressure head field $h$, the Darcy velocity $V$ [m.s$^{-1}$] is derived
according to equation (3) :

$$V(h,T) = K_H(h,T) . \nabla(h+z) \tag{3}$$


In the energy balance equation (2), the considered transfer processes are conduction through the
entire porous medium, convection by pore water flow, and latent heat exchanges when phase
changes occurs. In this heat transfer equation, $K_{T,eq}$ [J.m$^{-1}$.s$^{-1}$.K$^{-1}$] is the apparent thermal
conductivity of the porous medium, $\theta_{ice}$ [-] is the volumetric ice content, L [J.m$^{-3}$] is the latent heat
of fusion of ice, $C_{T,eq}$ [J.m$^{-3}$K$^{-1}$] is the equivalent heat capacity of the porous medium, and $C_{T,liquid}$
[J.m$^{-3}$K$^{-1}$] is the equivalent heat capacity of liquid water. In permaFoam these two coupled equations
are solved in 3D using the finite volumes method, with sequential operator splitting for handling the
couplings, Picard loops for dealing with the non-linearities, and a backward time scheme for
temporal discretization. A detailed description of the solver and can be found in Orgogozo et al.

157 (2023).

The numerical resolution of these coupled and highly non-linear equations, including stiff
fronts generated by freeze/thaw processes, at the space and time scales required for studying climate
change impacts on boreal watersheds, requires both robust algorithm and efficient use of high
performance computing means. This is the reason why permaFoam is developed within the
OpenFOAM framework, which allows benefiting from up-to-date and efficient numerical methods
for solving partial differential equations on last generation supercomputing facilities. Thanks to its



implementation in OpenFOAM, the permaFoam solver has demonstrated excellent parallel
performances on various supercomputer architectures, both in terms of large numerical domains (up
to 1 billion mesh points on the CALMIP Olympe supercomputer) and number of cores (16,000 on
the GENCI IRENE-ROME supercomputer) (Orgogozo et al., 2023).
According to preliminary numerical experiments (data not shown), for modelling Kulingdakan
watershed permafrost the use of a dual 2D simplified representation, with a 2D transect for
representing each slope of the watershed (like in Orgogozo et al., 2019), does not induced
significant loss of information compared to full 3D modelling. Meanwhile, 3D simulations are far
more costly from a computational prospect than 2D simulation (Orgogozo et al., 2023). Besides, the
use of 2D simulations allows considering lateral transfers (Sjöberg et al., 2016, Lamontagne-Hallé
et al., 2018, Hamm and Frampton, 2021, Jan 2022). Thus in this paper we use 2D numerical
domains, with climatic forcing as top boundary conditions (see section 2.3) and geothermal heat
flux and nil water flux as bottom boundary conditions. The initial conditions were obtained by 10
years of spin-up under current climatic conditions. These current climatic conditions were
represented by a synthetic year of climate forcing corresponding to the multi-annual means of the
1999-2014 observations. The starting conditions of this spin-up were the extracted from results of
the previous calculations (Orgogozo et al., 2019). The convergence criterion for the spin-up was the
active layer thickness inter-annual difference (annual variability less than 0.2%).

182       The numerical simulations provide the full 2D fields of physical quantities describing the
heat and water flow within the both SAS and NAS (two 2.5 km wide, 10 m thick slopes), including
both frozen and active layer in each slope. These included  soil temperature, pressure head, liquid
water content and ice content for each time step of saving (user defined, here each 6 months). In
addition, temperature, water content, ice content and evapotranspiration sink term are monitored at
hourly frequency throughout two vertical profiles located at mid-slope of SAS and NAS numerical
domains, using 61 virtual point probes distributed over the ten metres of the numerical domain
thickness. Finally, the infiltration and exfiltration water fluxes through the total soil surface are also
saved from the standard output at every time step.

191       This modelling set up is described in more details in  Supplementary material B.

***2.3 Soil surface conditions under climate change derived from CMIP6 scenarios***

193       In order to apply climate forcing that are representative of possible future trajectories, we
consider climate scenarios produced as a part of the Coupled Model Intercomparison Project Phase



6 (CMIP6) organized by the Intergovernmental Panel on Climate Change (IPCC) (Eyring et al., 2016), and in particular the so-called tier-1, key scenarios (O'Neill et al.,2016). These scenarios have been highlighted because of their relevance to scientific questions, the range of possible futures they cover, and their continuity with previous RCP scenarios (Representative Concentration Pathways, van Vuuren et al., 2011) published during CMIP5. We considered four CMIP6 scenarios, from the coldest to the hottest: SSP1-2.6, SSP2-4.5, SSP3-7.0 and SSP5-8.5. Among these scenario, the SSP2-4.5 is most often used in permafrost studies (e.g.: Karjalainen et al., 2019, Ramage et al., 2021, Hjort et al., 2022). For each of these scenarios, an ensemble of models has been run on different regions of the globe. The climate model output data were accessed via the IPCC Working Group I (IPCC-WGI) Interactive Atlas (Iturbide et al., 2021), February 2023 version, which provides the median (P50) of the ensemble of models for a selected output variable, region and scenarios. We used the air temperature and precipitation projections for the East Siberian region. The temporal evolution of their annual means between 2015 and 2100 have been summed to the multi-annual (1999-2014) yearly averages measured in Tura, for obtaining the local scenarios of climate change to be considered in this work (Fig. 2).

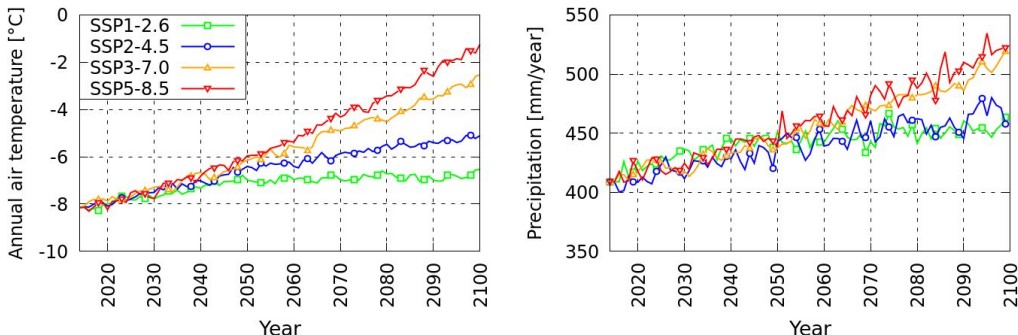

**Figure 2: Projections of air temperature and precipitation in Kulingdakan based on CMIP6 projections on the Eastern Siberia area.**

The projections show an increase in air temperature over the century, with a rate between +1.9 °C/100 yr (SSP1-2.6) and +7.8 °C/100 yr (SSP5-8.5), which is for every scenario higher than the global rate of increase (Fan et al., 2020). Annual precipitation could also change significantly, with a relative increase in 2100 of +12% (SSP1-2.6) to +29% (SSP5-8.5) compared to current value.



In order to translate these climate projections, that describe atmospheric conditions, into
suitable soil surface boundary conditions for cryohydrogeological simulations (water fluxes and
temperature at the soil surface, beneath snow and moss layer), a dedicated empirical procedure has
been developed. The goal is to set up a methodology for deriving soil surface temperature from air
temperature in the slopes of Kulingdakan watershed, based on the available observation data.
Indeed, soil temperature and air temperature may be significantly different in such a boreal forest
environment, due to the effects of understory (Zellweger et al., 2019, Haesen et al., 2021), moss
cover insulation (Blok et al., 2011, Cazaurang et al., 2023), the winter snowpack (Jan and Painter,
2020, Khani et al., 2023) and its interactions with vegetation (Dominé et al., 2022). This empirical,
site-specific procedure is detailed in  Supplementary material A, and it allows to build up a slope-
wise soil temperature estimates on the basis of air temperature and snow conditions. For water
fluxes, the simplest approximation has been adopted, assuming that the water flux at the top of the
soil is equal to the rain flux. For soil surface temperature estimate, we first used  a modified
temperature index approach (Braithwaite and Olesen, 1989, Hock 2003) for estimating snow water
equivalent, and then a multiple regression for deriving below moss, soil surface temperature from
air temperature, precipitation and snow water equivalent.
We chose a temperature index approach to simulate the snow water equivalent on the soil
surface because climate projections only provide air temperature and precipitation, whereas a more
advanced energy balance snowpack model requires additional information on wind, radiation, and
air humidity. To calibrate this temperature index model we first reconstructed the snow water
equivalent for the period 1999-2014 from the observed snow depth with the Multiple Snow Data
Assimilation System (MuSA) toolbox (Alonso-González et al., 2022) forced with ERA5 data
(Hersbach et al., 2020), fusing available snow depth observations with an ensemble of simulations
generated by the energy and mass balance model the Flexible Snow Model (Essery, 2015)). The
SWE model shows a good agreement with the MuSA reconstructions (Figure 3a), hence this model
was used to estimate SWE under future climate projections (Figure 3b).



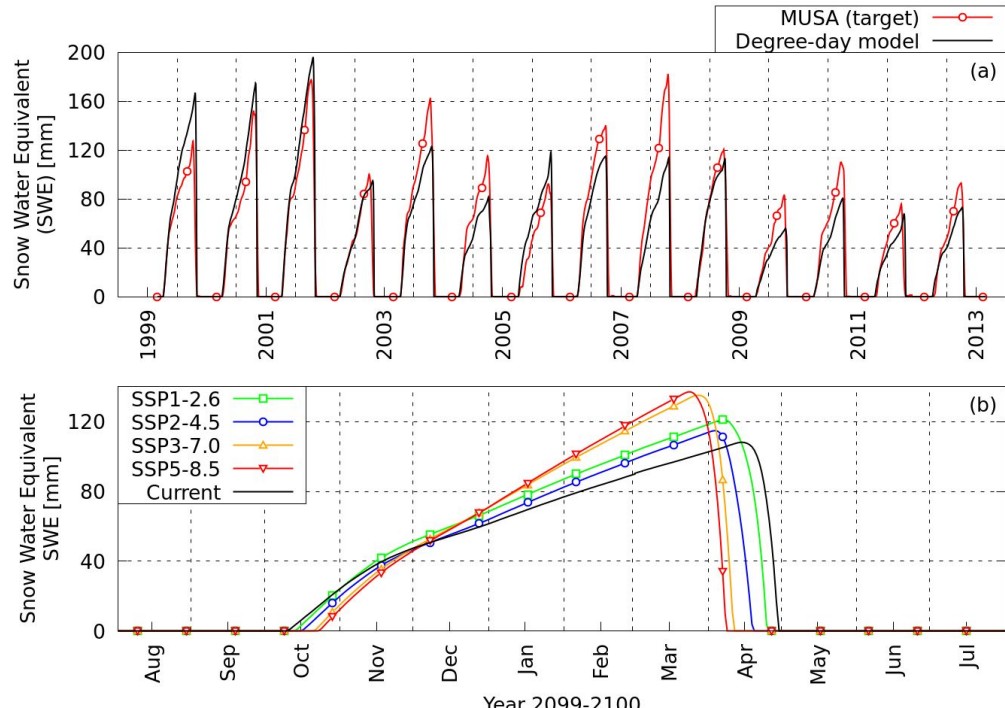

**Figure 3: Present snow model comparison with MuSA output (a) and projection at the end of the century (b)**

Then, we calibrate a multiple regression to derive soil surface temperature as a function of air temperature, while taking into account moss and snow layers insulating effect. Calibrations are performed with air temperature and precipitation data measurements and MuSA derived snow water equivalent between 1999 and 2014, and top soil (i.e., below moss) temperature measured in situ between 2003 and 2005. With this procedure, for each slope, an empirical transfer function that provides soil temperature estimates derived from air temperature and precipitation is obtained. These empirical transfer functions are in good agreement with the observation, as shown on Figure 4.

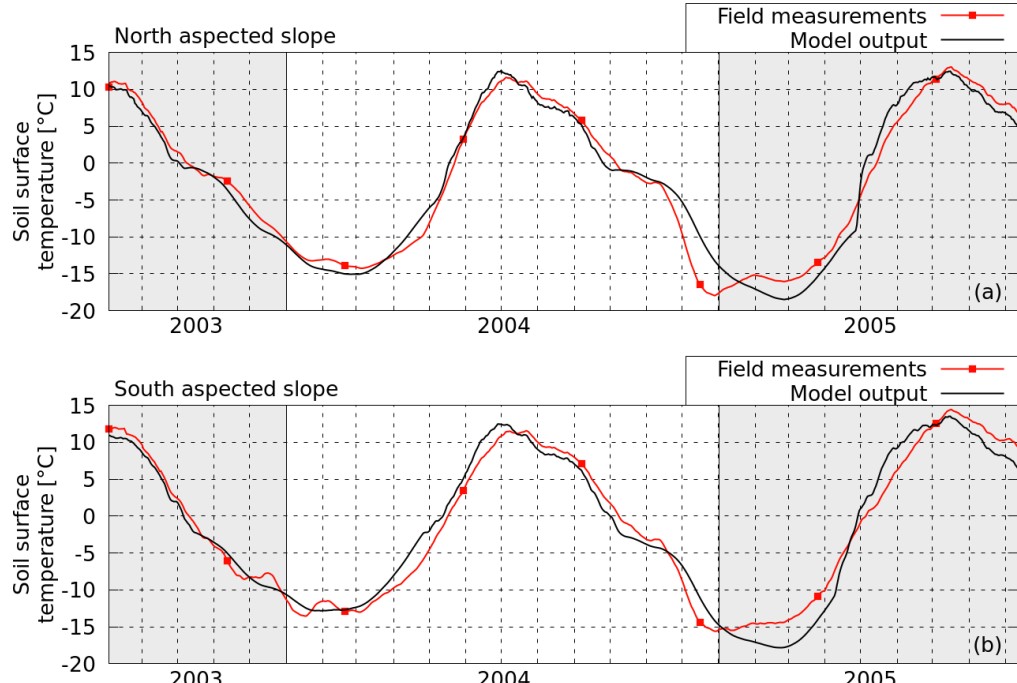

**Figure 4: Measurements and empirical transfer function estimates for soil surface temperature in present climatic conditions in NAS (a) and in SAS (b).**

Finally, these transfer functions are used to produce scenarios of soil surface temperature under climate change for the two slopes of the catchment (Fig. 5). This information is needed for building the soil surface boundary conditions of the hydrogeological simulations.





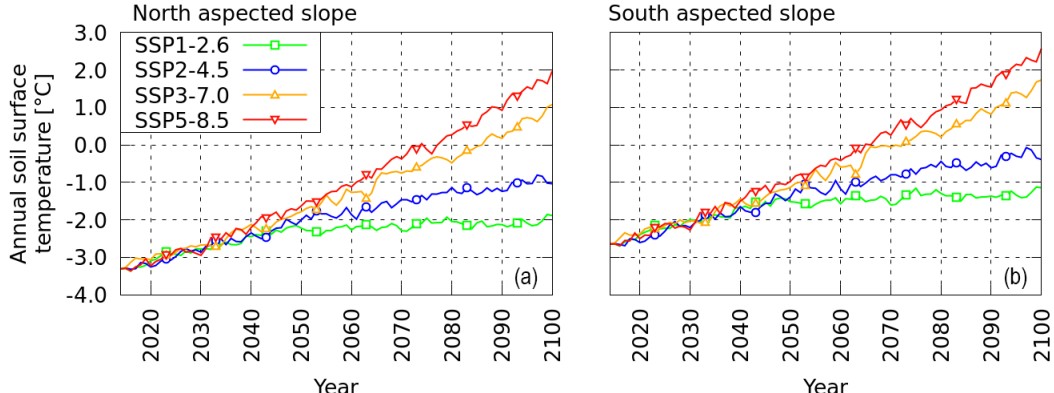

266

**Figure 5: Soil surface temperature projections over the century based on SSP scenarios obtained using transfer function described in Supplementary material A. Transfer function model estimation for soil surface temperature at present conditions for north aspected slope (a) and south aspected slope of the Kulingdakan watershed(b).**

271

The four projections based on the different SSP scenarios lead to an increase of the ground surface temperature from +1.4°C (SSP1-2.6) to +5.2°C (SSP5-8.5) between 2014 and 2100 (Fig. 5a and 5b). These rates of increase, roughly equivalent by extrapolation to +1.7°C/100 yrs (SSP1-2.6) and +5.9°C/100yrs (SSP-8.5), are lower than the observed increases in air temperature (+1.9°C/100yr for SSP1-2.6 and +7.8°C/100 yr for SSP5-8.5) due to the insulating effect of the snow cover and the vegetation layer, and also due to the thermal inertia of the soil column below the surface. One can note that for SSP3-7.0 and SSP5-8.5 scenarios, the mean annual soil surface temperature becomes positive around 2080. It must be emphasized that our empirical approach was based on parametrical fitting on observation data for estimating the transfer function between atmospheric forcing and soil surface temperature. As a result, no vegetation changes along climate change could be considered in this transfer function. Therefore, we focus on purely physical response of the catchment permafrost to climate change, taking into account vegetation impacts on permafrost dynamics at constant vegetation cover. Using an empirical transfer function for getting soil surface temperature signal from atmospheric conditions under climate change poses the problem of extrapolation, for instance in extreme hot weather conditions that may occur in the future, being unprecedented in the training period 1999-2014. However, performing mechanistic modelling of surface energy balance in extreme weather conditions under permafrost contexts was beyond the scope of this work.



### *2.4 High Performance Computing methodology*

Despite the use of the 2D assumption and the excellent parallel performance of the permaFoam solver, carrying out a mechanistic permafrost dynamics simulation at the scale of the catchment over almost a century remains a particular computational challenge. This section outlines some elements of the methodology and computing means used to meet this challenge.

The calculations are carried out on the IRENE JOLIOT-CURIE supercomputer operated by the French Alternative Energies and Atomic Energy Commission (CEA). This supercomputer offers, among other partitions, an AMD partition equipped with AMD Rome (Epyc) processors, with 64 computational cores each. OpenFOAM is used in this work only on CPUs with the MPI communication protocol. Since the mesh domain is composed by 525k cells for each slope (sufficient for convergence, see Supplementary material B), the number of MPI processes can be kept relatively low, with the use of 256 MPI-processes for each case treated here. Like most of fluid mechanics solvers based on finite volumes discretization, permaFoam exhibits a memory-bound nature in most of its operations, with low arithmetic intensity. Therefore, we adapted the use of the supercomputing infrastructure by depopulating the compute nodes by a factor of two (using only 32 computational cores out of the 64 cores available on each processor), thus largely broadening the bandwidth available for each MPI process. This operation reduced the computation time by almost a factor of two without requiring significantly higher CPU hour costs.

As a whole, the computational campaign required the use of 1.8 million CPU hours, generated almost 2TB of raw data and produced ~80k inodes, with a restitution time of approximately one month for each simulation (i.e.: for one scenario and for one slope).

## 3 Results

Post-processing the computed 2D fields of physical quantities describing the heat and water flow within the both SAS and NAS (two 2.5 km wide, 10 m thick slopes), including both frozen and active layer in each slope, a large wealth of data characterizing the considered virtual permafrost dynamics is obtained (Supplementary material C), and below, only key features of the centennial evolution under climate change are presented.





### 3.1 Trends in soil temperatures


Soil temperature at different depth is one of the key variable for characterizing permafrost
dynamics. The multi-annual trends induced by climate warming of the mean annual soil
temperature between 2014 and 2100 at 3 depths (10 cm, 1 m and 5 m below the surface) are
illustrated in Figure 6.

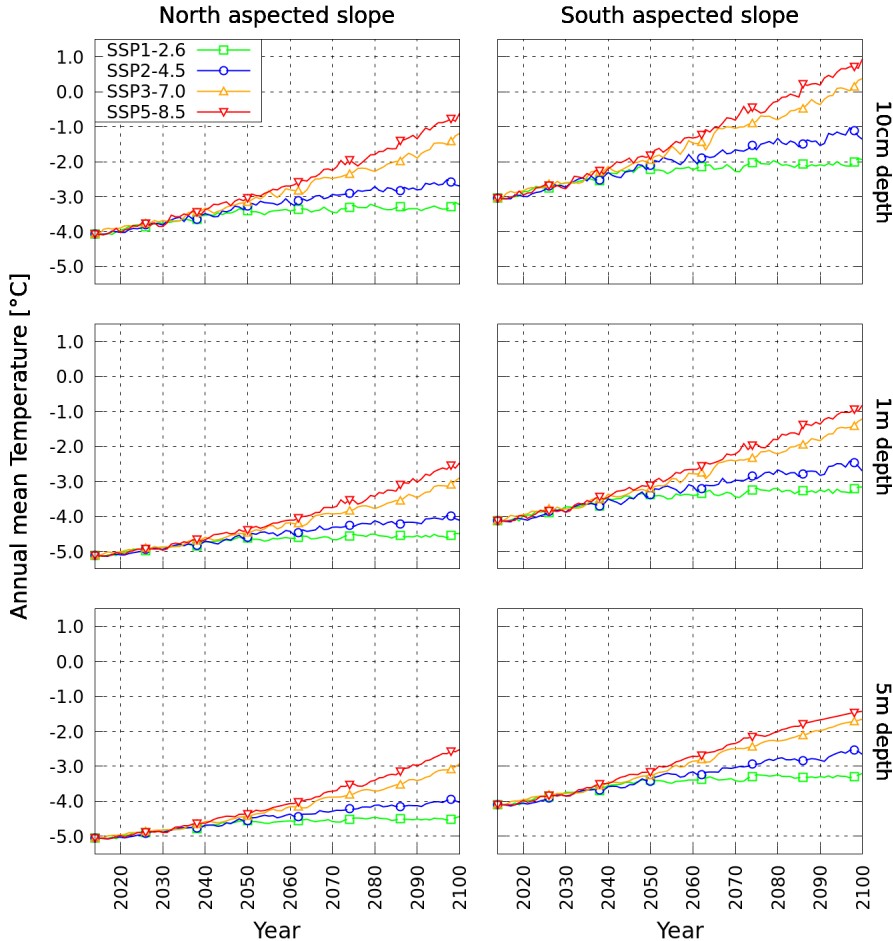


**Figure 6: Mean annual temperature evolution at 10cm,1m and 5m under the surface for each**
**scenario and slope considered.**



In both slopes, the soil temperature experiences significant increase down to 5 m depth, for
all climate warming scenario considered. The annual mean soil temperature becomes even positive
close to the surface in the SAS for the two hottest scenarios, by 2085 with SSP5-8.5 and by 2095
with SSP3-7.0. Meanwhile, for the medium scenario SSP2-4.5 and for the coldest scenario SSP1-
2.6, the mean annual soil temperature stay everywhere negative until 2100. The warming is more
intensive in the SAS than in the NAS, and, as expected, the amplitude of soil warming decreases
with depth. In SAS, at 10 cm depth the temperature rise between current conditions and the year
2100 is 1 °C for the SSP1-2.6 scenario and 4 °C for the SSP5-8.5 scenario, while at 5 m depth, the
temperature rises are 0.9 and 2.7 °C, respectively. In NAS, at 10 cm depth the temperature rise
between current conditions and the year 2100 is of 0.8 °C for the SSP1-2.6 scenario and of 3.4 °C
for the SSP5-8.5 scenario, while at 5 m depth, the temperature rises are 0.6 and 2.5°C respectively.
It should be noted that, for both slopes, the vertical gradient of temperature in 2100 is higher in
scenario SSP5-8.5 than in scenario SSP1-2.6. This indicates a stronger thermal non-equilibrium
under more intense warming. For instance, the difference of temperature between 10 cm and 5 m
depth is 2.3 °C in SAS and of 1.9 °C in NAS for scenario SSP5-8.5, while it is 1.2 °C in both SAS
and  NAS for the SSP1-2.6 scenario. In order to provide insight into the thermal equilibrium state of
the soil columns in each slope in 2100, the vertical temperature profiles for this year are plotted for
each scenario in Figure 7. For further investigating the distance from equilibrium state in each
slope, an additional numerical experiment has been performed by simulating 30 years more at the
2100 climatic conditions, and the obtained vertical temperature profiles are shown in Figure 7.



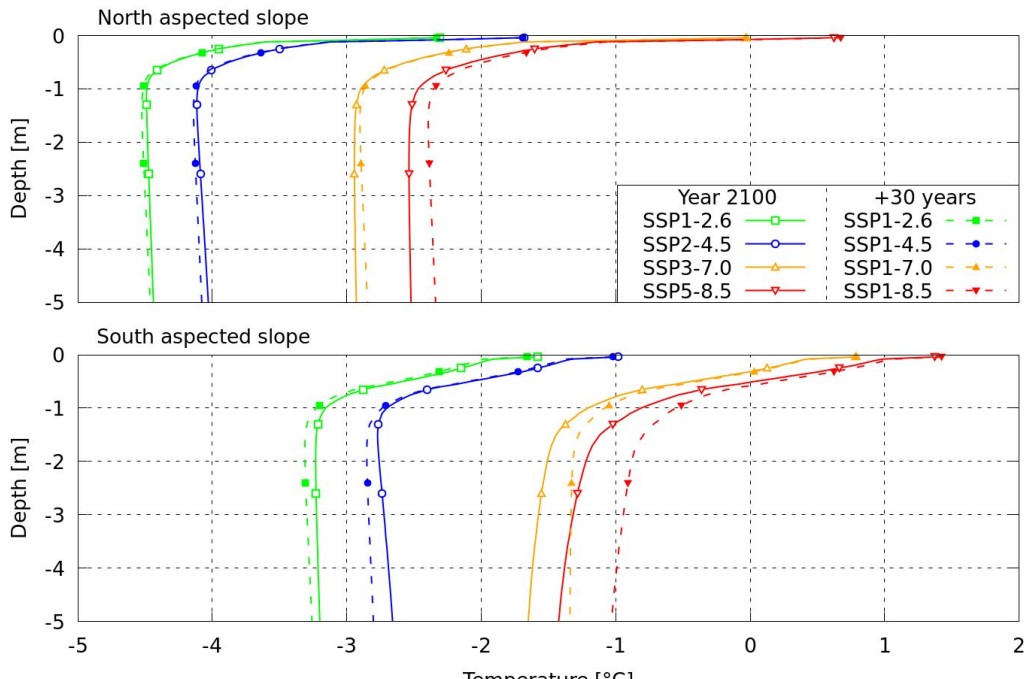


**Figure 7: Annual mean temperature profiles in 2100 and after 30 years of additional cycling of the climatic forcing of this last year.**


Considering the soil temperature profiles in 2100, two regions may be distinguished: the first meter with steep positive vertical gradients (the soil surface is warmer than the bottom of the active layer), and a deeper region, with smoother vertical thermal gradients, either slightly negative (SSP1-2.6 and SSP2-4.5 in NAS and SAS), almost nil (SSP3-7.0 and SSP5-8.5 in NAS) or positive (SSP3-7.0 and SSP5-8.5 in SAS). When comparing these profiles with those obtained with 30 additional years of modelling in constant '2100' climatic conditions, we observe important differences in both slopes for scenario SSP5-8.5, and also for scenario SSP3-7.0 and scenario SSP2-4.5 in SAS.

361

### 3.2 Active layer thickness evolution

Numerical simulations give access to soil temperature at various depth. From soil temperature profile, the maximum depth with a positive temperature may be computed at each time



step. The maximum thawed depth obtained over a year defines the active layer thickness (ALT) of
this year. Active layer thickness has been computed for each scenario and each year and is plotted
for both NAS and SAS in Figure 8.

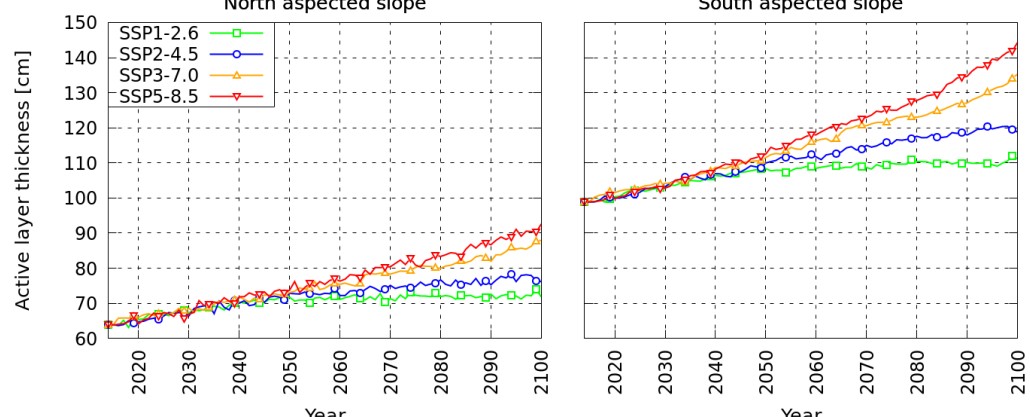

**Figure 8: Active layer thickness temporal evolution on North (left) and South (right) aspect
Slope of the Kulingdakan watershed obtained from permaFoam simulations under different
SSP scenarios.**

For both slopes, an increase in active layer thickness is observed between 2014 and 2100 in every
scenario, with a more important thickening in SAS than in NAS. SSP1-2.6 leads to an increase of
+12.3 cm/+12% for SAS and of +7.9 cm / +12% for NAS, while SSP5-8.5 leads to a more dramatic
increase of +45 cm / +46% for SAS and of +29 cm / +45% for NAS. In the first half of the century,
the behavior of active layer thickness does not differ significantly between scenarios, with a
thickening rate in ALT of about +3.3 mm/year (±21%) in SAS and of +2.5 mm/year (±19%) in
NAS. However, in the second half of the century (2050-2100), different scenarios lead to very
different active layer thickness evolution dynamics. For SSP1-2.6, the thickening rate is rather
small, with a rate of +0.52 mm/year for SAS and +0.25 mm/year for NAS, while for SSP5-8.5
scenario, the thickening rate rises to +5.8 mm/year for SAS and +3.5 mm/year for NAS. By the end
of the simulated period, these thickening rates show no diminishing trend in SAS, suggesting that
the dynamic thermal equilibrium is not reached in the active layer. For illustrating this, Figure 9
shows the active layer thickness evolution for 30 years of additional simulations while keeping the
climatic conditions of 2100 for each scenario.




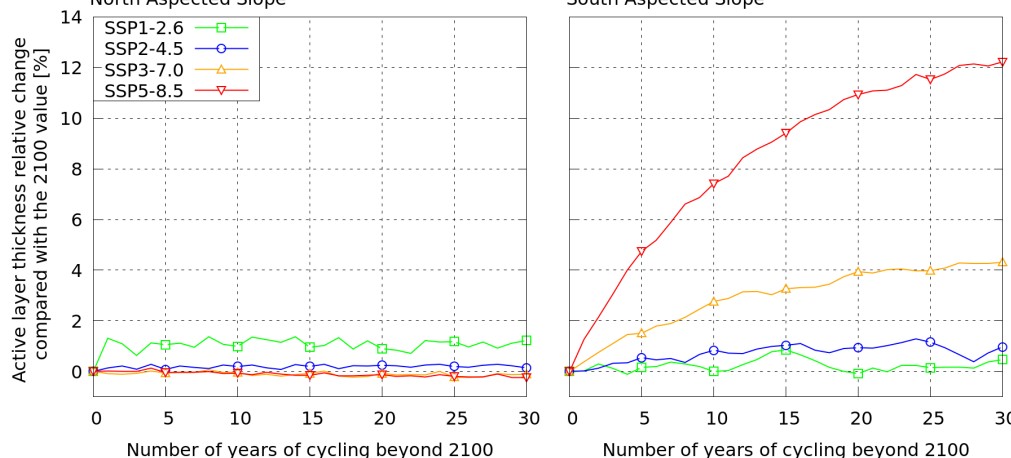

**Figure 9: Relative change in active layer thickness compared with the year 2100 over 30 years**
**of spin-up of the 2100 climatic conditions.**

Overall, the active layer is not far from thermal equilibrium in both slope for the coldest (SSP1-2.6)
and medium (SSP2-4.5) climatic scenarios. However, when considering the hottest SSP5-8.5
scenario, an important thermal inertia effect appears in SAS, with   an additional active layer
thickness increase over these 30 years of +12.6 % compared to the 2100 value, i.e. an increase of
+18 cm. This additional change in active layer thickness brings the resulting change compared to
2014 value to +63 cm (+64%) for the hottest scenario.






### *3.3 Trends in soil moisture*
Soil moisture content experienced less important changes than thermal regime under the
considered climate change scenario. For illustrating the soil moisture evolution near the surface, the
total water, liquid water and ice volumetric contents have been averaged over the first 2 m of the
soil for each slopes, and their 2014-2100 evolution have been plotted on Figure 10 for the four
climatic scenarios. Regardless of the scenario, there is no significant evolution of total water content
in the first two meters of soil in NAS, and the only noticeable change is the increase in proportion
of liquid water  (+15 % in SSP1-2.6, +24 % in SSP2-4.5, +49% in SSP3-7.0, +60 % in SSP5-8.5),
suggesting an increase in the amount of liquid water available for vegetation.  In SAS however, the
first two meters of the soil exhibited a slight but detectable diminishing of total water content by
2100 (-5 % in SSP1-2.6 and SSP2-4.5, -10% in SSP3-7.0, -11 % in SSP5-8.5). On the other hand
the proportion of liquid water over ice increases (+7 % in SSP1-2.6, +16 % in SSP2-4.5, +38% in
SSP3-7.0, +51 % in SSP5-8.5). Therefore, in SAS slope, climate warming may result in no
significant changes in the amount of liquid water available for vegetation. It should be emphasized
that the presented partitioning between liquid water and ice is based on the mean annual quantities.
This provides considerably smaller proportion of liquid water compared to that in the end of the
active season (second half of September), when active layer is at its maximum thickness.




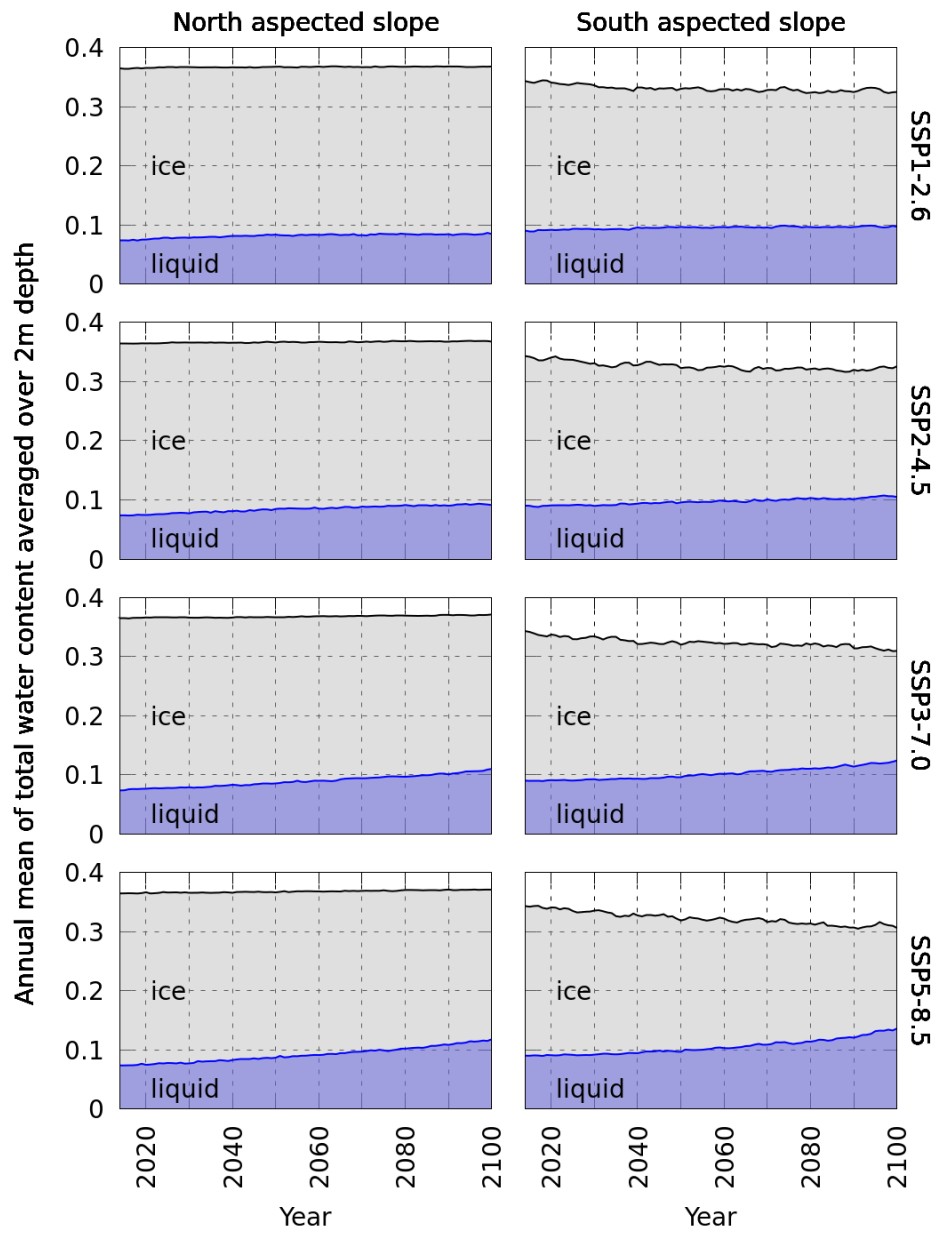

**Figure 10: Annual mean of total water content [m$^3$ of water / m$^3$ of soil], liquid water content and ice content averaged over 2m depth in different climate projections.**








In order to investigate the local variation of moisture content in the rooting zone and in the
active layers of each slopes, the vertical profiles of mean annual total water content have been
plotted on Figure 11 for current climatic conditions and for year 2100 under SSP1-2.6, SSP2-4.5,
SSP3-7.0 and SSP5-8.5 scenarios. The water profiles do not change significantly in the highly
porous organic horizon in both slopes. In the mineral horizon, the behavior of SAS and NAS get
more contrasted, due to downward vertical moisture gradients (and thus upward water movements)
in NAS and upward vertical moisture gradients (and thus downward water movements) in SAS. In
NAS, the only evolution with climate change is a thickening of the zone with a downward vertical
moisture gradient (i.e., an upward water flux) alongside the thickening of the active layer, with no
significant changes of the gradient itself. Meanwhile, in SAS, alongside with the thickening of the
zone with water movements (i.e. moisture gradients) that comes with active layer thickening,
significant changes of the upward moisture gradients are expected to occur: the hotter the scenario,
the steeper the gradients, and thus the stronger the downward water fluxes.






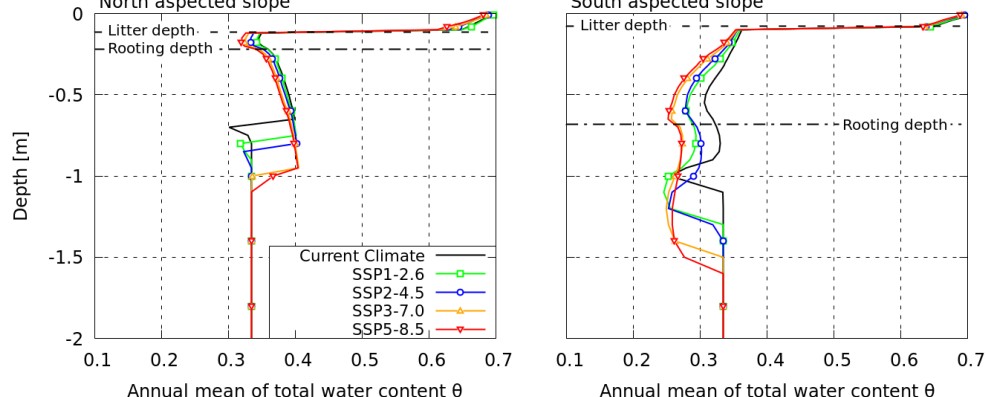

**Figure 11: 2m-depth profiles of annual mean of total water content [m$^3$ of water / m$^3$ of soil] in 2100: projections compared to current state.**

*3.4 Water fluxes*

The water fluxes also significantly change along with climate change in both slopes for every scenario. Evapotranspiration is the most important component of the hydrological budget in Kulingdakan. Focusing on this dominant component, Figure 12 presents the centennial evolution of evapotranspiration in both slopes and of precipitation for the fourth considered climate change scenarios. A significant increase of evapotranspiration is simulated in all cases, with an increase between +18 mm/+5% (SSP1-2.6) and +94 mm/+26% (SSP5-8.5) in SAS, and between +4 0mm/+11% and +100 mm/+29% in NAS. The increase of the evapotranspiration fluxes in Kulingdakan are closely correlated to the increase of precipitation, with similar increase rates for both slopes.



450

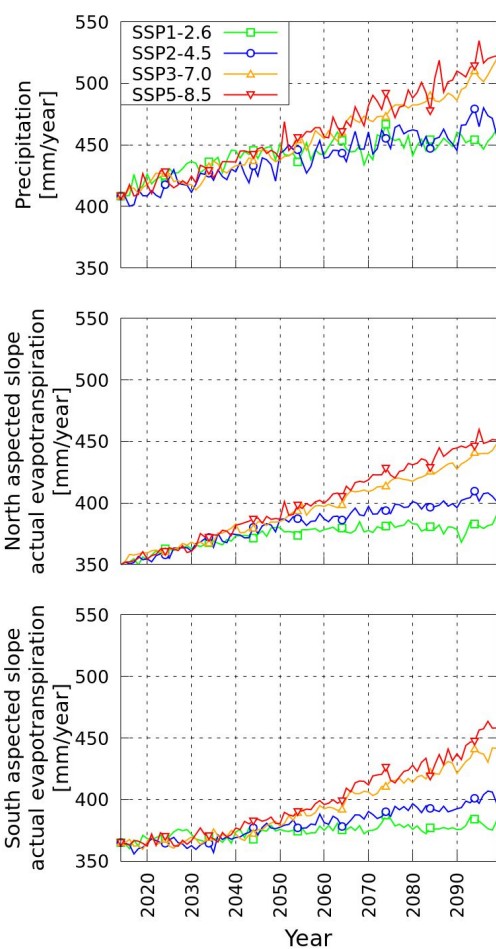

451

**Figure 12: Precipitation and actual evapotranspiration evolution over the century**

Once again, the evolution is globally similar among scenarios until 2050, with significant divergences appearing only between 2050 and 2100.

455





## 4 Discussion

The numerical results obtained by mechanistical modelling of heat and water transfer within the permafrost and active layer of Kulingdakan document the physical response to be expected by this catchment under climate change, with soil warming (Figure 6) and active layer thickening (Figure 8) in all climate scenarios. An important spatial variability of this thermal response is identified, in relation with the aspect of the slopes, which stems from sizable contrast in vegetation cover, hydrologic and thermal state and active layer dynamics as currently observed between the two slopes (Prokushkin et al. 2007). Indeed, since the NAS is wetter, its thermal inertia is more important due to the largest amount of latent heat that must be provided to thaw and warm its soils, compared to the drier soils of the SAS. This difference in moisture content is largely due to the difference in their tree cover biomass and physiology, i.e. the deeper root layer in SAS compared to NAS induces more intensive evapotranspiration in the former, both under current (Orgogozo et al., 2019) and future climate conditions. Note that this contrast tends to diminish with climate warming (Figure 12). Nonetheless, according to the performed simulations, the SAS will stay drier than NAS with climate warming (Figure 10). The structure of water fluxes within the active layer, with an upward flux to the thinner, close to the surface root layer in NAS and a downward flux toward the bottom of the thicker root layer in SAS is also preserved under climate change, with an intensification of the fluxes in SAS under the hottest scenarios (Figure 11). Further, the thicker moss layer in NAS is likely to dampen more efficiently the effect of changes in the climatic conditions on soil compared to the thinner one of the SAS. Because our modelling takes into account the root water uptake mechanistically (Orgogozo et al., 2023) and the low vegetation insulating effect empirically (Supplementary material A), the warming of the soil and the thickening of the active layer under climate change is significantly more pronounced in SAS than in NAS. This persistent local spatial variability in permafrost dynamics reflects the prominent role of micro-climatic condition in responses of forest environments to climate change that has been demonstrated recently in the literature (Zellweger et al., 2020). It must be emphasized that all these numerical results have been obtained considering the vegetation in its present state. Strong local variability of the vegetation cover depending on the permafrost conditions in Kulingdakan catchment (Orgogozo et al., 2019) and in a broader perspective in the entire Arctic (Oehri et al., 2022) is consistent with important interplay between vegetation evolution under climate change (e.g. Vitasse et al. 2009, 2011, Rew et al., 2020) and permafrost pattern, which has not been explicitly considered in this





study. At the centennial time scale, changes in the tree growth rate, the forest fire frequency or the
nature of the vegetation cover may exert important impacts on permafrost conditions (Cable et al.,
2016; Fedorov et al., 2019; Rew et al., 2020; Li et al., 2021; Heijmans et al., 2022). Meanwhile,
without belittling these probably important and complex interactions between vegetation dynamics
and permafrost dynamics, this study shows that important impacts of climate change are to be
expected on permafrost dynamics of the forested continuous permafrost area even at steady
vegetation cover. We noted that the more intense the climate change, the more pronounced these
thermal responses. For instance, under the hottest SSP5-8.5 scenario, a maximum evolution of the
active layer thickness is +45 cm/+46% for SAS and +28 cm/+44% for NAS, while in the medium
SSP2-4.5 scenario, an increase of +20 cm/+20% for SAS and of +12 cm/+19% for NAS is
anticipated.

498       To produce a broader geographical context of the simulated active layer thickening at the
scale of a small catchment, a comparison of these centennial evolutions under climate change with
large geographical coverage is performed in Figure 13.

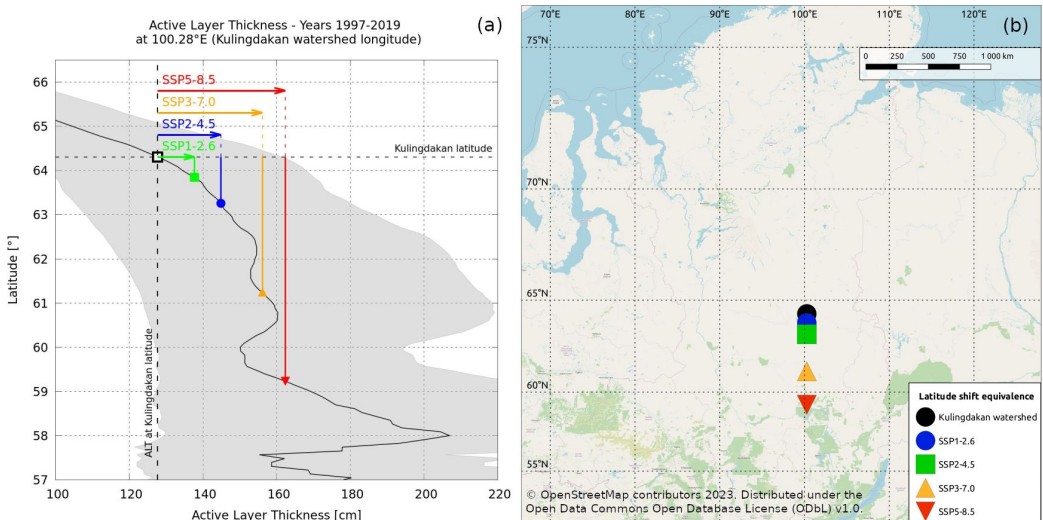

**Figure 13: (a) Equivalence between simulated active layer thickening by 2100 under climate**
**change (SAS and NAS average) and southward latitudinal shift in current climatic conditions**
**(1997-2019). – latitudinal trend (black line - average over a 1°lat. × 1°long. polygon) and**
**envelops (in grey - min/max over year within the same polygon) extracted from**
**Permafrost_cci (Obu et al., 2021). (b) Representation of the latitudinal southward shift**
**equivalent to each climate scenario's active layer thickening on the regional map.**



The simulated thickening of the active layer, averaged over both slopes of Kulingdakan, are depicted as southward latitudinal shifts along the meridian passing by Kulingdakan, i.e. with a pure North-South translation along the 100.28 °E (Fig. 13). The latitudinal evolution of the active layer thickness along the current meridian is computed based on the permafrost_CCI dataset (Obu et al., 2021), by averaging the value of the multi-annual mean of active layer thickness for the 1997-2019 period over a polygon of 1° latitude by 1° of longitude centered on the considered meridian and browsing the latitudes 67°N and 57°N. It can be seen that, in the hottest scenario SSP5-8.5, the active layer thickening would correspond to a 560 km southward shift, while in the medium scenario SSP2-4.5, it would correspond to a 120 km southward shift.

Under a permanently changing climatic context, an important question is the state of thermal equilibrium versus non-equilibrium of the permafrost (Obu et al., 2019): is the climate change induced warming slow enough so that permafrost may be considered at every time close to the thermal equilibrium with climatic conditions, or on the contrary, the transient effects dominate the thermal dynamics of permafrost under climate change? The simulation results of this work provide information for characterizing the degree of thermal equilibrium of the continuous permafrost, in a forested study site under various scenarios of climate change. First of all, we emphasize that, since the bottom thermal boundary condition in our modelling is geothermal heat flux (Duchkov et al., 1997), the assumption of overall thermal equilibrium in depth (< 10 m) in the hundreds of meter of thick permafrost of the Putorana plateau (Pokrovsky et al., 2005) is implicitly made. Meanwhile, the temperature profiles shown in Figure 7 demonstrate that under this assumption the thermal equilibrium state of the first 10 m of soil in 2100 depends on both the climate change scenario and the slope aspect. In NAS, thermal equilibrium of the first 10m of soil is achieved by 2100 in every climate scenario, with only a slight shift between 2100 and (2100 +30) conditions in SSP5-8.5 scenario. Besides, with sub-zero vertical thermal gradients in each scenario, only small heat exchanges between surface and depth occur. On the contrary, by 2100 in SAS strong thermal non-equilibrium is encountered in the two hottest scenarios, SSP3-7.0 and SSP5-8.5 (Figures 7 and 8). Under these scenarios, sizable evolutions of temperature profiles are expected to occur between 2100 and 2100+30. Moreover, for these two hottest scenarios, the vertical thermal gradient between 1 and 10m depth are clearly positive (considering an upward vertical axis), which implies an ongoing heat flux from the surface to the depth. In this case, the permafrost is warming below 10m, at a rate that we implicitly assume to be small enough so that it does not modify the total amount of heat stored within this deep permafrost. As such, in scenarios SSP3-7.0 and SSP5-8.5, the climate



warming clearly induces transient warming of the permafrost in depth (below 10m) in the south aspect slopes of the Kulingdakan watershed. One could note a slightly decreasing trends in the soil temperature under scenarios SSP1-2.6 and SSP2-4.5. This is due to inter-annual variabilities in both precipitation and air temperature in CMIP6 projections (Figure 2). Therefore, the year 2100, which is repeated over 30 cycles to assess the equilibrium state of the permafrost, may offer different conditions from those observed in the previous decade 2090-2100. For example, in SSP2-4.5, the last decade experiences an important annual precipitation peak, up to 475mm/year, centered around 2095, before a decreasing trend on the second part of the decade ending up to a precipitation of 410mm/year projected in 2100. This results for the year 2100 in a decrease of the snow cover insulating effect in winter, and thus a cooling of soil surface temperature (Fig. 5), compared to the conditions encountered in the previous decade,

Overall, results of the present study may be used to improve our understanding of the climate warming-related changes in the wide areas of boreal forest on continuous permafrost, with strong implications for continental surfaces (Revich et al., 2022), ecosystems (Wang and Liu 2022) and element cycles (Schuur et al., 2022) there, and related global consequences and feedbacks. The use of mechanistic modelling, although computationally costly, is capable of providing quantitative information for feeding these research fields. This approach will be applied in other environmentally monitored boreal watershed in the near future, in order to numerically characterize the physical response of permafrost to climate change in various representative permafrost context, for instance in Northern Sweden (Auda et al., 2023) and Western Siberia (Cazaurang et al., 2023).

## 5 Conclusion

Four main conclusions that could be drawn from this numerical study are the following:

- All climate change scenarios trigger significant soil warming (+1.5°C in SAS and +1°C in NAS under SSP2-4.5 scenario at 1 meter depth according to the presented simulations) and an increase in the active layers thickness (+20 cm/+20% in SAS and +12 cm/+19% in NAS under SSP2-4.5 scenario) in both slopes of the Kulingdakan watershed.

- For all climate change scenarios, the combination of soil warming and precipitation increase leads to an important increase in evapotranspiration in both slopes (+34 mm/+9% in SAS and +52 mm/+15% in NAS under SSP2-4.5 scenario). Meanwhile, the mean annual soil moisture decreases only slightly in NAS (-2.3% under SSP2-4.5 scenario, averaged over the 22 cm of rooting depth),



but it is more pronounced in SAS (-5,6% in NAS under SSP2-4.5 scenario, averaged over the 68 cm of rooting depth).

- Important spatial variability observed in the Kulingdakan watershed illustrate the key role of meso-climatic conditions and small-scale geomorphological contrasts in permafrost response to climate warming

- Under the two hottest scenarios of climate change SSP3-7.0 and SSP5-8.5, the near-surface permafrost of the SAS of the Kulingdakan watershed are in non-equilibrium thermal state in 2100, and further investigation is needed to assess whether or not the permafrost underneath 10m depth will be close to thermal equilibrium in this region. This advocates the need of developing non-equilibrium modelling approaches for regional and global permafrost modelling under climate change.

The approach developed in this study can be applied to other high-latitude permafrost-affected catchments, provided that necessary information on current thermal and hydrological parameters of soil as well as vegetation coverage are available.

**Competing interests**

The contact author has declared that none of the authors has any competing interests.

**Acknowledgments**

This work as been funded by the French National Research Agency ANR (grant n° ANR-19 CE46-0003-01), and benefited from access to the supercomputers of CALMIP (project p12166) and GENCI (project A0140410794, TGCC). Oleg Pokrovsky is grateful for support from the TSU Development Programme PRIORITY – 2030 and PEPR "Peace". Anatoly Prokushkin is supported by State Assignment no. 0287-2021-0008. Esteban Alonso Gonzalez is supported by the European Space Agency through the Climate Change Initiative postdoctoral grant.



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
