# Peer review of "Siberia: quantitative assessment with a mechanistic modelling approach"

_EGUsphere, 2023_

## Author Comment (AC2)

**FIRST PART: ANSWERS TO REFEREE 2**

[Figure]

Figure R2.1: Vertical profiles of vertical temperature gradient in the middle of the North Aspected Slope (left) and in the middle of the South Aspected Slope (right), under current climate and in 2100 for four climate change scenarios.

**SECOND PART: SELF-MOTIVATED CORRECTION**

Comparison between ill-results and correct results:

*Figure 6: Mean annual temperature evolution at 10cm, 1m, 5m and 10m under the surface for each scenario and slope considered.*

*ILL:*                                                                    *CORRECT:*

[Figure]

- Left: Soil temperature evolutions with ill parameterization (with soil temperature amplitude limitation to the amplitude of monthly mean top soil temperature under current climate);
- Right: Soil temperature evolutions with correct parameterization (without soil temperature amplitude limitation).

*Figure 7: Annual mean temperature profiles in 2100 and after 30 years of additional cycling of the climatic forcing of this last year.*

ILL :

[Figure]

CORRECT :

[Figure]

*Figure 8: Active layer thickness temporal evolution on North (left) and South (right) aspect Slope of the Kulingdakan watershed obtained from permaFoam simulations under different SSP scenarios.*

ILL:

[Figure]

- Above: Active layer thickness evolutions with ill parameterization (with soil temperature amplitude limitation to the amplitude of monthly mean top soil temperature under current climate);

CORRECT:

[Figure]

- Above: Active layer thickness evolutions with correct parameterization (without soil temperature amplitude limitation).

*Figure 9: Relative change in active layer thickness compared with the year 2100 over 30 years of spin-up of the 2100 climatic conditions.*

ILL :

[Figure]

CORRECT :

[Figure]

*Figure 10: Annual mean of total water content [m 3 of water / m 3 of soil], liquid water content and ice content averaged over 2m depth in different climate projections.*

*ILL:* *CORRECT:*

[Figure]

- Left: Water and ice contents evolutions with ill parameterization (with soil temperature amplitude limitation to the amplitude of monthly mean top soil temperature under current climate);
- Right: Water and ice contents evolutions with correct parameterization (without soil temperature amplitude limitation).

*Figure 11: 2m-depth profiles of annual mean of total water content [m 3 of water / m 3 of soil] in 2100: projections compared to current state.*

*ILL:*

[Figure]

- Above: 2m-depth profiles of annual mean of total water content [m$^3$ of water / m$^3$ of soil] in 2100: projections compared to current state – ill parameterization (with soil temperature amplitude limitation to the amplitude of monthly mean top soil temperature under current climate);

*CORRECT*:

[Figure]

Above: 2m-depth profiles of annual mean of total water content [m$^3$ of water / m$^3$ of soil] in 2100: projections compared to current state – correct parameterization (without soil temperature amplitude limitation).

*Figure 12: Precipitation and actual evapotranspiration evolution over the century*

*ILL:*                                                                                                      *CORRECT:*

[Figure]

- Left: Actual Evapotranspiration evolutions with ill parameterization (with soil temperature amplitude limitation to the amplitude of monthly mean top soil temperature under current climate);
- Right: Actual Evapotranspiration evolutions with correct parameterization (without soil temperature amplitude limitation).

*Figure 13: (a) Equivalence between simulated active layer thickening by 2100 under climate change (SAS and NAS average) and southward latitudinal shift in current climatic conditions (1997-2019). – latitudinal trend (black line - average over a 1°lat. × 1°long. polygon) and envelops (in grey - min/max over year within the same polygon) extracted from Permafrost_cci (Obu et al., 2021). (b) Representation of the latitudinal southward shift equivalent to each climate scenario's active layer thickening on the regional map.*

ILL:

[Figure]

- Above: Equivalent ALT-Latitude shift based on Permafrost_cci ALT data (Obu et al.,2021) - ill parameterization (with soil temperature amplitude limitation to the amplitude of monthly mean top soil temperature under current climate);

CORRECT:

[Figure]

- Above: Equivalent ALT-Latitude shift based on Permafrost_cci ALT data (Obu et al.,2021) - correct parameterization (without soil temperature amplitude limitation).

| Latidudinal shift compared to Kulingdakan Latitude (64.31°N) | ILL parametrization | CORRECT parametrization |
|---|---|---|
| SSP1-2.6 | -0.46° / 51km | -0.45° / 50km |
| SSP2-4.5 | -1.06° / 118km | -1.17° / 130km |
| SSP3-7.0 | -3.10° / 345km | -5.20° / 578km |
| SSP5-8.5 | -5.06° / 563km | -5.64° / 628km |

*Supplementary material C: tables compiling the main variables change between present conditions and 2100 for the four climate scenarios considered in this paper (SSP1-2.6, SSP2-4.5, SSP3-7.0, SSP5-8.5) for North Aspected Slope (Table C1) and South Aspected Slope (Table C2)*

| ILL: Variables (NAS) | Annual value in present climate | Change from present values in projections to 2100 | | | |
|---|---|---|---|---|---|
| | | SSP1-2.6 | SSP2-4.5 | SSP3-7.0 | SSP5-8.5 |
| Air temperature | -8.2°C | +1.6°C | +3.0°C | +5.6°C | +6.9°C |
| Yearly precipitations | 408mm | +56mm / +14% | +49mm / +12% | +111mm / +27% | +115mm / +28% |
| Maximum snow water equivalent | 108mm | +7mm / +6% | +13mm / +12% | +27mm / +25% | +29mm / +27% |
| Snow season extent | 202days | -6days | -8days | -14days | -17days |
| Soil surface temperature | -3.3°C | +1.4°C | +2.3°C | +4.3°C | +5.2°C |
| Soil temperature (10cm depth) | -4.1°C | +0.9°C | +1.4°C | +2.9°C | +3.4°C |
| Soil temperature (1m depth) | -5.12°C | +0.6°C | +1.0°C | +2.2°C | +2.6°C |
| Soil temperature (5m depth) | -5.06°C | +0.6°C | +1.0°C | +2.2°C | +2.5°C |
| Soil temperature (10m depth) | -4.9°C | +0.6°C | +1.0°C | +2.0°C | +2.5°C |
| Active layer thickness | 64cm | +7.8cm +12% | +11.9cm +19% | +23.9cm +37% | +28.2cm +44% |
| Total water content (averaged over root layer) | 0.510 | +1.7x10$^{-4}$ +0.0% | -1.2x10$^{-2}$ -2.3% | -1.7x10$^{-2}$ -3.3% | -2.4x10$^{-2}$ -4.7% |
| Liquid water content (averaged over root layer) | 0.197 | +1.2x10$^{-2}$ +6.3% | +1.4x10$^{-2}$ +7% | +2.8x10$^{-2}$ +14.1% | +3.3x10$^{-2}$ +16.5% |
| Ice water content (averaged over root layer) | 0.313 | -1.2x10$^{-3}$ -3.9% | -2.6x10$^{-2}$ -8.1% | -4.5x10$^{-2}$ -14.3% | -5.6x10$^{-2}$ -18% |
| Total water content (averaged over 0-2m) | 0.365 | +3.2x10$^{-3}$ +0.9% | +3.0x10$^{-3}$ +0.8% | +6.7x10$^{-3}$ +1.8% | +6.3x10$^{-3}$ +1.7% |
| Liquid water content (averaged over 0-2m) | 0.074 | +1.1x10$^{-2}$ +14.7% | +1.8x10$^{-2}$ +23.8% | +3.6x10$^{-2}$ +49.3% | +4.4x10$^{-2}$ +60.1% |
| Ice water content (averaged over 0-2m) | 0.291 | -7.7x10$^{-3}$ -2.7% | -1.5x10$^{-2}$ -5.0% | -3.0x10$^{-2}$ -10.2% | -3.8x10$^{-2}$ -13.1% |
| Actual evapotranspiration | 350mm | +40mm / +11% | +52mm / +15% | +98mm /+28% | +100mm / +29% |

*Table C1: main variables changes simulated for the four climate projections in 2100 for north aspected slope  - ill parameterization (with soil temperature amplitude limitation to the amplitude of monthly mean top soil temperature under current climate);*

| *CORRECT:* Variables (NAS) | Annual value in present climate | Change from present values in projections to 2100 | | | |
|---|---|---|---|---|---|
| | | SSP1-2.6 | SSP2-4.5 | SSP3-7.0 | SSP5-8.5 |
| Air temperature | -8.2°C | +1.6°C | +3.0°C | +5.6°C | +6.9°C |
| Yearly precipitations | 408mm | +56mm / +14% | +49mm / +12% | +111mm / +27% | +115mm / +28% |
| Maximum snow water equivalent | 108mm | +7mm / +6% | +13mm / +12% | +27mm / +25% | +29mm / +27% |
| Snow season extent | 202days | -6days | -8days | -14days | -17days |
| Soil surface temperature | -3.3°C | +1.4°C | +2.3°C | +4.3°C | +5.2°C |
| Soil temperature (10cm depth) | -4.6°C | +1.2°C | +1.9°C | +3.7°C | +4.4°C |
| Soil temperature (1m depth) | -5.6°C | +1.0°C | +1.5°C | +2.9°C | +3.4°C |
| Soil temperature (5m depth) | -5.6°C | +1.0°C | +1.5°C | +2.8°C | +3.2°C |
| Soil temperature (10m depth) | -5.5°C | +0.9°C | +1.5°C | +2.7°C | +3.2°C |
| Active layer thickness | 63cm | +8.8cm +14% | +14.5cm +23% | +30.9cm +49% | +38.5cm +61% |
| Total water content (averaged over root layer) | 0.510 | $1.1 \times 10^{-4}$ +0.0% | $-1.2 \times 10^{-2}$ -2.3% | $-1.7 \times 10^{-2}$ -3.4% | $-2.4 \times 10^{-2}$ -4.7% |
| Liquid water content (averaged over root layer) | 0.198 | $1.2 \times 10^{-2}$ +5.9% | $1.3 \times 10^{-2}$ +6.5% | $2.7 \times 10^{-2}$ +13.8% | $3.2 \times 10^{-2}$ +16.3% |
| Ice water content (averaged over root layer) | 0.312 | $-1.2 \times 10^{-2}$ -3.7% | $-2.5 \times 10^{-2}$ -7.9% | $-4.4 \times 10^{-2}$ -14.2% | $-5.6 \times 10^{-2}$ -18.0% |
| Total water content (averaged over 0-2m) | 0.364 | $+3.5 \times 10^{-3}$ +1.0% | $+3.9 \times 10^{-3}$ +1.1% | $+9.4 \times 10^{-3}$ +2.6% | $+9.3 \times 10^{-3}$ +2.6% |
| Liquid water content (averaged over 0-2m) | 0.072 | $+1.2 \times 10^{-2}$ +17.3% | $+2.0 \times 10^{-2}$ +28.4% | $+4.5 \times 10^{-2}$ +62.4% | $+5.6 \times 10^{-2}$ +77.8% |
| Ice water content (averaged over 0-2m) | 0.292 | $-8.9 \times 10\text{-}03$ -3.1% | $-1.7 \times 10^{-2}$ -5.7% | $-3.6 \times 10^{-2}$ -12.2% | $-4.7 \times 10^{-2}$ -16.0% |
| Actual evapotranspiration | 351mm | +35mm/+10% | +51mm/+14% | +108mm/+31% | +123mm/+35% |

*Table C1: main variables changes simulated for the four climate projections in 2100 for north aspected slope - correct parameterization (without soil temperature amplitude limitation).*

| *ILL:* | Annual value in present climate | Change from present values in projections to 2100 | | | |
|---|---|---|---|---|---|
| Variables (SAS) | | SSP1-2.6 | SSP2-4.5 | SSP3-7.0 | SSP5-8.5 |
| Air temperature | -8.2°C | +1.6°C | +3.0°C | +5.6°C | +6.9°C |
| Yearly precipitations | 408mm | +56mm/+14% | +49mm / +12% | +111mm / +27% | +115mm / +28% |
| Maximum snow water equivalent | 108mm | +7mm / +6% | +13mm / +12% | +27mm / +25% | +29mm / +27% |
| Snow season extent | 202days | -6days | -8days | -14days | -17days |
| Soil surface temperature | -2.6°C | +1.5°C | +2.3°C | +4.4°C | +5.2°C |
| Soil temperature (10cm depth) | -3.1°C | +1.1°C | +1.7°C | +3.4°C | +4.0°C |
| Soil temperature (1m depth) | -4.15°C | +1.0°C | +1.5°C | +2.9°C | +3.3°C |
| Soil temperature (5m depth) | -4.11°C | +0.9°C | +1.5°C | +2.4°C | +2.7°C |
| Soil temperature (10m depth) | -4.0°C | +0.9°C | +1.5°C | +2.3°C | +2.5°C |
| Active layer thickness | 99cm | +13cm +13% | +20.0cm +20% | +36.3cm +37% | +45.2cm +46% |
| Total water content (averaged over root layer) | 0.375 | $-1.6 \times 10^{-2}$ -4.3% | $-2.1 \times 10^{-2}$ -5.6% | $-3.2 \times 10^{-2}$ -8.5% | $-3.7 \times 10^{-2}$ -9.7% |
| Liquid water content (averaged over root layer) | 0.153 | $+1.1 \times 10^{-3}$ +0.7% | $+3.5 \times 10^{-3}$ +2.3% | $+1.2 \times 10^{-2}$ +8.0% | $+1.5 \times 10^{-2}$ +9.8% |
| Ice water content (averaged over root layer) | 0.222 | $-1.7 \times 10^{-2}$ -7.7% | $-2.4 \times 10^{-2}$ -11.0% | $-4.4 \times 10^{-2}$ -19.9% | $-5.1 \times 10^{-2}$ -23.1% |
| Total water content (averaged over 0-2m) | 0.343 | $-1.8 \times 10^{-2}$ -5.4% | $-1.7 \times 10^{-2}$ -5.0% | $-3.4 \times 10^{-2}$ -9.8% | $-3.7 \times 10^{-2}$ -10.8% |
| Liquid water content (averaged over 0-2m) | 0.090 | $+6.6 \times 10^{-3}$ +7.3% | $+1.5 \times 10^{-2}$ +16.2% | $+3.4 \times 10^{-2}$ +37.7% | $+4.6 \times 10^{-2}$ +50.7% |
| Ice water content (averaged over 0-2m) | 0.253 | $-2.5 \times 10^{-2}$ -9.9% | $-3.2 \times 10^{-2}$ -12.6% | $-6.8 \times 10^{-2}$ -26.8% | $-8.3 \times 10^{-2}$ -32.7% |
| Actual evapotranspiration | 364mm | +18mm/+5% | +34mm/+9% | +76mm/+21% | +94mm/+26% |

*Table C2: main variables changes simulated for the four climate projections in 2100 for south aspected slope - ill parameterization (with soil temperature amplitude limitation to the amplitude of monthly mean top soil temperature under current climate);*

| *CORRECT:* | Annual value in present climate | Change from present values in projections to 2100 | | | |
|---|---|---|---|---|---|
| Variables (SAS) | | SSP1-2.6 | SSP2-4.5 | SSP3-7.0 | SSP5-8.5 |
| Air temperature | -8.2°C | +1.6°C | +3.0°C | +5.6°C | +6.9°C |
| Yearly precipitations | 408mm | +56mm/+14% | +49mm / +12% | +111mm / +27% | +115mm / +28% |
| Maximum snow water equivalent | 108mm | +7mm / +6% | +13mm / +12% | +27mm / +25% | +29mm / +27% |
| Snow season extent | 202days | -6days | -8days | -14days | -17days |
| Soil surface temperature | -2.6°C | +1.5°C | +2.3°C | +4.4°C | +5.2°C |
| Soil temperature (10cm depth) | -3.3°C | +1.4°C | +2.1°C | +4.2°C | +5.0°C |
| Soil temperature (1m depth) | -4.4°C | +1.3°C | +1.8°C | +3.5°C | +4.0°C |
| Soil temperature (5m depth) | -4.4°C | +1.2°C | +1.8°C | +2.9°C | +3.1°C |
| Soil temperature (10m depth) | -4.3°C | +1.2°C | +1.8°C | +2.7°C | +2.9°C |
| Active layer thickness | 100cm | +12.5cm +13% | +22.6cm +23% | +46.5cm +47% | +65.1cm +65% |
| Total water content (averaged over root layer) | 0.375 | $-1.8 \times 10^{-2}$ -4.9% | $-2.2 \times 10^{-2}$ -6.0% | $-3.2 \times 10^{-2}$ -8.6% | $-3.5 \times 10^{-2}$ -9.4% |
| Liquid water content (averaged over root layer) | 0.152 | $+7.6 \times 10^{-4}$ +0.5% | $+3.9 \times 10^{-3}$ +2.5% | $+1.2 \times 10^{-2}$ +8.2% | $+1.6 \times 10^{-2}$ +10.3% |
| Ice water content (averaged over root layer) | 0.223 | $-1.9 \times 10^{-2}$ -8.5% | $-2.6 \times 10^{-2}$ -11.8% | $-4.5 \times 10^{-2}$ -20.1% | $-5.1 \times 10^{-2}$ -22.9% |
| Total water content (averaged over 0-2m) | 0.339 | $-1.6 \times 10^{-2}$ -4.6% | $-1.8 \times 10^{-2}$ -5.4% | $-3.2 \times 10^{-2}$ -9.5% | $-3.4 \times 10^{-2}$ -9.9% |
| Liquid water content (averaged over 0-2m) | 0.089 | $+8.2 \times 10^{-3}$ +9.2% | $+1.8 \times 10^{-2}$ +20.4% | $+4.4 \times 10^{-2}$ +49.8% | $+6.4 \times 10^{-2}$ +72.4% |
| Ice water content (averaged over 0-2m) | 0.250 | $-2.4 \times 10^{-2}$ -9.6% | $-3.6 \times 10^{-2}$ -14.5% | $-7.7 \times 10^{-2}$ -30.6% | $-9.8 \times 10^{-2}$ -39.1% |
| Actual evapotranspiration | 361mm | +19mm/+5% | +37mm/+10% | +82mm/+23% | +108mm/+30% |

*Table C2: main variables changes simulated for the four climate projections in 2100 for south aspected slope - correct parameterization (without soil temperature amplitude limitation).*

---

## Author Response (AR1)

**REVIEWER 1**

**1. Methods**

The methods section is fairly extensive and contains a lot of information about the site as well as the model. However, from subsection 2.3 onward, it also contains results that are better placed in the result section. It would be good if the authors could clearly separate methods and results to make it easier for the reader to follow.
*The proposed re-organization has been applied (l 296-333).*

**Specific comments**

1. L53-54: It would be helpful to add a number and reference here as to how much of the permafrost area is covered by boreal forest.
*About 55 % of the permafrost affected area is covered by boreal forest (Stuenzi et al., 2021). This information has been added to the updated manuscript (l 57).*

2. L75-78: Here the objectives of the paper can be highlighted more clearly. Naming the actual quantities that are being investigated (soil temperature, moisture, active layer thickness, southward shift) can help forming specific objectives that then are answered in a very concise bullet point list in the conclusion section of the current paper. Linking these two will make it easier for the reader to understand the full picture.
*The proposed improvement has been applied (l 97-103 and l 617-619, l 624).*

3. L101: Also provide an average active layer thickness either for each, NAS and SAS, or overall. This can also be done after the NAS and SAS had been introduced in one of the following sentences.
*The proposed improvement has been applied (l 123, l 128-129).*

4. L103-105: The description of the catchment is a bit abstract. A figure with a map of the catchment(s) would be extremely helpful here. It could be part of Figure 1.
*The proposed improvement has been applied (see Figure 1, l 133-136).*

5. L168-171: This sentence is very long and a bit confusing, please clarify.
*The sentence has been splitted and rewritten for the sake of clarity (l 186-189).*

6. L178-188: A plot with the forcing data described in this sentence would be a valuable addition to the methods. It can be part of the supplement.
*This plot has been added to the Supplementary material A (l 32-42), and references to it have been added in the body of the text (l 195-196, l 232).*

7. L200 and throughout the entire manuscript: "coldest" and "hottest" scenarios do not adequately describe the conditions in the SSP scenarios. Terms like "high-end forcing pathway" (for SSP5-8.5) and "low-forcing sustainable pathway" (for SSP1-2.6) are more adequate.
*The proposed terminology has been used in the updated manuscript.*

8. L207-209: I do not fully understand this sentence. What is meant by "the means haven been summed"?
*What we call « current climatic conditions » consist in a multi-annually averaged daily climatic*

*forcing dataset producing a virtual representative year, constructed from observations available between 1999 and 2014. The multi-annual average is made day by day for each meteorological variable (air temperature, precipitation) for all the years of the considered observation period to build this synthetic dataset. For instance, the precipitation of the first January of the synthetic year is the average of the precipitations of the first Januaries from 01/01/1999 to 01/01/2014. Then this current climatic conditions synthetic year is used for performing the initial spin-up of the permafrost simulations. To build the climate scenarios, we add to this synthetic year the trends of yearly averaged air temperature and precipitation changes from 2015 to 2100, trends obtained from CMIP6 scenarios available for East Siberian region (IPCC). In this way we derived the used climatic projections for air temperature and precipitation in Kulingdakan (Fig. 2). For instance, the air temperature signal for year 2050 for a given climatic scenario is the sum of the air temperature of the current climatic conditions synthetic year and of the offset computed from the considered climate scenario. We rephrased the sentence for being more precise and clear (l 226-234). Explanations have also been added in supplementary material A for clarification (l32-42).*

9. L215: Is the rate +1.9°C/100yr interpolated to a full 100 year period? From my understanding you are only looking at the time frame 2014-2100, which is not exactly 100 years. Please clarify. *These rates are calculated for the 2014-2100 timeframe. For instance, the +1.9°C/100 yr corresponds to a +1.65 °C/87 yr increase for the considered simulation period. We chose to rescale the rates to century time scale for the sake of readability. This has been explained in the updated manuscript (l 240-241).*

10. L216: What is the global temperature increase rate? Please provide and cite this information. *Global air temperature increases over the 21st century from Fan et al.,2020 are given below and have been added to the manuscript (l 241-244).*
*SSP1-2.6 : +1.18 °C/100 yr*
*SSP2-4.5 : +3.22 °C/100 yr*
*SSP3-7.0 : +5.50 °C/100 yr*
*SSP5-8.5 : +7.20 °C/100 yr*
*Fan, X., Duan, Q., Shen, C., Wu, Y., and Xing, C., Global surface air temperatures in CMIP6: historical performance and future changes, Environmental Research Letters, vol. 15, no. 10, IOP, 2020. doi:10.1088/1748-9326/abb051.*

11. L238 and throughout the methods section: It is not entirely clear to me with which temporal resolution you are working. The snow cover plots seem to be interpolated over the year using monthly values (Fig 3.) but for soil surface temperature you are providing annual means. Please clarify this in the method section.
*For snow cover estimation and soil surface temperature, the solver works at daily timestep. The smoothness in the SWE curves presented in figure 3b is due to the smooth shape of atmospheric condition signals used as input for the model (the synthetic year of current climatic conditions, see answer to point 8). However, for vizualisation purpose when addressing the century timescale, we use a rolling annual average of the displayed quantity. This provides the climatic trend obtained, while avoiding the display of intra-annual variability, which would make the figure unreadable on a century-scale plot. It should be mentioned that soil surface temperature and water fluxes are provided to the permaFoam solver with a daily frequency. However, the permaFoam solver makes use of adaptative timestepping, with here minimal time steps of 1 second. OpenFOAM proposes natively the linear interpolation of the daily boundary conditions to the time step sequence required*

*by the permafrost simulations. We added explanations regarding the use of daily climatic input data in the method section (l 233-234, l 276).*

12. L286-287: This is still part of the methods section but you are introducing an important discussion point, which would be better placed in the discussion section
*The proposed re-organization has been be applied, with these sentences moved to the Discussion section (l 528-532).*

13. Section 2.4: I am unsure about how useful this information is for the main text of the paper. It is certainly interesting but might be better placed in the supplementary information as it is distracting from the main text. Also, I am curious about the amount of energy that was required to perform this modeling using 1.8 million CPU hours. Is it possible to convert this computational effort into an energy equivalent, offering a rough estimate of the energy required for permafrost modeling and comparing it to a more relatable context? For instance: The energy consumed by our numerical modeling process is equivalent to the energy required to power a typical 4-person household for [X] days/months/years, assuming an average household energy consumption rate of [Y] kWh per day/month/year." Information like this will be increasingly important with computer codes increasing in complexity and ever higher performing supercomputers.
*The section regarding High Performance Computing requirements and methodologies has been moved to the Supplementary Material B (l 203-236). Regarding the energy consumption associated with the performed simulations, we propose the following estimate. On IRENE-ROME supercomputer, the power consumption is estimated to 5,02734 W/core (personnal communication from the operating team). Thus the energy consumption of our 1.8 millions of CPU hours simulation campain could be roughly estimated to 9MWh. For comparing this consumption with the one of a typical 4-person household in the European Union, we propose to consider the final energy consumption in households (all end uses, including water heating, space heating and cooling, cooking and electrical appliance) available in the Eurostat database for the year 2021 (1584677 terajoules\*, equal to 440188055 MWh). Then, this total consumption may be divided by the census population in 2021 (445649015 inhabitants\*\*) to obtain an estimate of the average energy consumption per person and per year in the European Union (0,987 MWh/person/year). According to this estimate, the energy consumed by our numerical modeling process is equivalent to the energy required to power a typical 4-person household for about 27 months. This comparison has been added to the Supplementary material B (l 223-236).*
*\* https://doi.org/10.2908/NRG_D_HHQ*
*\*\* https://doi.org/10.2908/CENS_21AG*

14. Figure 7: Is it possible to add the current soil temperature profile to these plots?
*The current soil temperature profiles (one for each slope) have been added (l 366-367).*

15. L383-387: This feels like an afterthought. It would be helpful if the authors would introduce the idea of simulating 30 extra years to test for thermal equilibrium section to explain what it is used for. Then this part could also be shortened and becomes more clear.
*The proposed improvement have been applied (l 361-364).*

16. Section 3.3: The first part of section 3.3 and Figure 10 are hard to understand for me. The quantity of "total water content averaged over the first 2m" is very abstract and does not provide much information. Could the authors please either highlight which processes are depending on changes in the total water content (vegetation is mentioned here, but for this aspect Figure 11 would

be enough) or potentially remove part of this section and Figure 10? Figure 11 provides more information at a glance and is more easily understandable.

*Figure 10 presents temporal evolution until 2100 of the mean annual total water content averaged on the first two meters of the soil – this thickness of the considered surface soil layer is chosen so that it encompasses all the area with water content evolution under the considered climate change scenarios (see vertical profiles in Figure 11). The two main points of this figure are the following :*
*- illustrating the evolutions of moisture content of surface soils in the slopes of Kulingdakan watershed, stable in NAS and slightly decreasing in SAS ;*
*- illustrating the evolutions in the partition of total soil water between liquid water and ice. These information are important for heat and water transfers in the soil, due to the couplings and non-linearities between these transfers. For instance, decreasing total water content induce decreasing soil thermal inertia, while decreasing share of ice vs liquid water induce a decrease in apparent thermal conductivity. These information are also important for vegetation dynamics, since vegetation needs soil water uptakes for transpiration, and may only uptake liquid water. These elements have been added to section 'Trends in soil moisture', now section 3.4 (l 428-430 and l 438-443).*

17. L428-431: What is meant by "upward water movements" here? Figure 11 is very illustrative and important, but I cannot follow its description very well. Partly because both SAS and NAS are being mentioned in alternating sequence and partly because the processes leading towards the shift in soil moisture are not well described. Hence, the second part of section 3.3 can be extended and a bit more information can be provided on the processes causing the soil moisture gradients to change.

*Figure 11 presents the comparison of vertical profiles of the annual mean of total water content under current climatic conditions and in 2100 for the four considered climate scenarios. The processes driving these evolutions of vertical moisture profiles are complex, involving coupled and non-linear heat and water transfers as well as changing evapotranspiration fluxes. Then the comment of Figure 11 is largely descriptive: according to the simulations, we observe changes in the moisture gradients within the soil. Then these moisture gradients can be interpreted in terms of water movements according to the generalized Darcy's law. The second part of section 'Trends in soil moisture', now section 3.4, has been rewritten so that the limit of the proposed interpretations appear in a clearer way (l 454-456 and l 459).*

18. L467 and onwards: While evaporation is discussed briefly here, I think it deserves more attention as it is a crucial variable of future climate change in the Arctic (see e.g., Clark et al. 2023). Its potential effect on soil temperatures and runoff would be interesting to discuss in a boreal forest context.

*For now in permaFoam, evapotranspiration is assumed to be solely constituted by transpiration, while the evaporation within the soil is neglected (Orgogozo et al., 2019). This assumption is made in the context of the study of boreal forest areas, in which transpiration largely dominates over evaporation in the hydrological budget (e.g., Park et al., 2021). Meanwhile, evaporation may be dominant in tundra environments (Clark et al., 2023), and may increase in the future in forested environments under climate change. Since considering soil evaporation would add another coupling between heat and water transfers through exchanges of latent heat, it could directly affect soil temperature evolution. These points will be the scope fo future modelling works, and a related preliminary discussion has been added to this manuscript (l 533-541).*
*Clark, J.A., Tape, K.D. and Young-Robertson, J.M., 2023. Quantifying evapotranspiration from*

*dominant Arctic vegetation types using lysimeters. Ecohydrology, 16(1), p.e2484.*
*Park, H., Tanoue, M., Sugimoto, A., Ichiyanagi, K., Iwahana, G., & Hiyama, T. (2021). Quantitative separation of precipitation and permafrost waters used for evapotranspiration in a boreal forest: A numerical study using tracer model. Journal of Geophysical Research: Biogeosciences, 126, e2021JG006645. https://doi.org/10.1029/2021JG006645*

19. Figure 13: Panel a is is very dense in information and an attempt is being made at describing what the gray shaded area and the black line are supposed to represent, but the methodology behind this approach is described very briefly and raises questions such as 1) is the 1°x1° polygon comparable in terms of vegetation type to the catchment in question? 2) Does it make sense to look at the entire time frame from 1997-2019 when looking at ALT? Active layer deepening over the last years is likely to skew the representation of the current state of the Active Layer. An average over the e.g., last 5 years might be more representative of the current state of the active layer. Since the dataset is only available until 2019, the years 2015-2019 might be suitable in this case.

*First, it should be emphasized that in boreal forest environments active layer thickness can significantly vary even at scales way smaller than 1°: exposition, vegetation cover, soil profile, topography can affect the active layer. In the Kulingdakan catchment for instance, strong differences exist between the two slopes of the watershed, at the km scale. These spatial heterogeneities can be seen at the permafrost_cci product resolution (1 pixel = 926.63m). Moreover, interannual variability also results in strong changes in ALT. In the current state of Fig 13(a), the black line describes the multi-annual (1997-2019) temporal average of the spatial average of the active layer thickness over a 1°-1° polygon centered on a moving latitude ; the gray shaded area represents the min/max obtained for this spatial average during the considered period. We can observe that the interannual variabilty can reach 40% of the average value. Therefore, in a given 1°-1° polygon at a given latitude, active layer thickness is strongly variable both spatially and temporally. However, in order to propose a southward shift equivalent estimation based on a latitudinal variability, the 1°-1° polygon was considered big enough to smooth the small-scale inhomogeneities (~km) and small enough to capture the latidunal effect, including biome transitions (~hundreds of km) (see Anisimov et al., 2015 for vegetation zone). Regarding the timeframe considered (1997-2019), the choice of the maximum data available was made in order to reduce the interannual variability effect. Meanwhile, we agree with the referee that this timeframe does not represent the most accurately the current state of the active layer, in a context of a rapid climate change. An updated figure considering only the 2017-2021 timeframe for current climate state has been proposed , along with a more detailled description of the methodology used for producing it (l 542-545, l 560-565).*

*Anisimov, O. A., Zhiltcova, Y. L., and Razzhivin, V. Y., Predictive modeling of plant productivity in the Russian Arctic using satellite data, Izvestiya Atmospheric and Oceanic Physics, vol. 51, no. 9, Springer, pp. 1051–1059, 2015. doi:10.1134/S0001433815090042.*

**REVIEWER 2**

*Major comments :*

1 - The study lacks a comparison of important thermos-hydrologic components against observations such as active layer thickness, evapotranspiration, and watershed runoff. The authors compared SWE and surface temperature against observations, but that is not sufficient to understand if the model can accurately simulate permafrost conditions, and importantly ALT.

*Numerical modelling of permafrost conditions under current climate conditions in the study site has been presented in previous papers (Orgogozo et. al, 2019, Orgogozo et al., 2023). These simulation results has been obtained with permaFoam, the same cryohydrogeological solver used for building the centennial projections presented in the submitted manuscript. These results were in good agreement with the available observations of active layer thickness and soil temperature profiles. This point have been better highlighted in the updated manuscript (l 116-117).*

2 - The authors focus on boreal forests, but vegetation dynamics are not included. This looks to me incomplete and inconsistent. Vegetations play a critical role in regulating ALT and ET. Is this due to model limitations?

*The role of vegetation in regulating ALT and ET is taken into account in permaFoam through a mechanistic approach, by computing a sink term in the root layer accounting for tree water uptake on the basis of the potential evapotranspiration and of the soil water content (Orgogozo, 2015, Orgogozo et al., 2019). The main conclusion of the previously published numerical study of permafrost conditions under current climate conditions is that the transpiration by vegetation is a key parameter for active layer dynamics in the study site (Orgogozo et al., 2019). Then vegetation is taken into account in the used modelling approach. What is not taken into account is the evolution of vegetation cover in response to climate change (e.g.: change in root layer thickness). Coupling a vegetation dynamics model with the cryohydrogeological model used here would allow to study the impact of the climate warming-induced changes of the vegetation cover on permafrost conditions. This is beyond the scope of the present study and will be the focus of future works. This point have been better highlighted in the updated manuscript (l 283-286).*

3 - The abstract needs to be revised. In lines 24-25, the authors mention quantifying soil temperature, soil moisture, active layer thickness, and water fluxes, however, the abstract reports an analysis of active layer and evapotranspiration only. In a modeling paper, more explicit quantification should be provided about important thermo-hydrologic components.

*The active layer thickness is a variable that strongly integrates heat and water transfers in permafrost affected soils. This is the reason why we put it forward in the report of results in the abstract. For keeping the abstract concise enough we will not include analysis for all the variable considered in the study, but we will rephrase it for highlighting the integrative nature of ALT (l 28-29).*

4 - There are many grammatical mistakes in the manuscripts that require careful consideration.

*A proofreading service has been requested to review the updated version of the manuscript.*

5 - The domain depth is 10 m, in such shallow domains, the bottom boundary conditions could impact the results as the surface thermal signal could penetrate deeper than 10 m, especially during projection. How sensitive are these results to the bottom boundary conditions?

*In this study we do not impose a fixed temperature at the bottom of the domain, but a fixed gradient equal to the geothermal flux. Thus this bottom boundary condition should not be surface conditions / climate change dependent. In order to illustrate this, the figure below shows the computed vertical profile of mean annual vertical temperature gradient in the middle of both slopes, for current climatic conditions and in 2100 for the four considered scenarios of climate change.*

[Figure]

*Figure R2.1: Vertical profiles of vertical temperature gradient in the middle of the North Aspected Slope (left) and in the middle of the South Aspected Slope (right), under current climate and in 2100 for four climate change scenarios.*

*One can see that as expected thermal gradients are much more important close to the surface than at 10 m depth. Besides, the thermal gradients vary only slightly below 5 m depth. Then assuming a fixed thermal gradient equal to the geothermal flux at the bottom of the modelling domain at 10 m depth is a reasonable approximation. It does not mean that temperature does not vary at 10 m depth. In order to illustrate the changes of temperature at the bottom of the domain we added plots for 10 m depth in Figure 6 (l 342-343).*

**Minor comments :**

L18: thermos-hydric? This is not a common terminology in the Arctic hydrology literature.
*The 'thermo-hydric' adjective has been changed into 'thermal and hydrological' (l 18).*

L27: I assume the plus sign in +46% and +29% indicates increases compared to the current climate. Please clarify it.
*The interpretation of the sentence is correct. The sentence has been clarified (l 28).*

L28: this line is confusing without proper context.
*The idea here is to propose a 'space for time' illustrative approach: to which southward spatial shift*

*in current climatic conditions would correspond the future increase of active layer thickness simulated under climate change projections? The sentence has been reformulated for making this idea clearer (l 29-30).*

L43: please provide the area percent covered by boreal.
*This information has been added (l 57).*

L120: Fig1. Please add a DEM image of the watershed, if available.
*We added to Fig1. a vizualisation of the DEM of the Kulingdakan watershed (l 135-136).*

L133: please clarify what "freeze-thaw of pore water" means here. Does it not freeze/thaw dry soil?
*Around 0°C phase changes occur only for water, not for the solid part of the soil. The permaFoam solver does take this into account in its description of variably saturated and variably frozen porous media as four phase (soil grains, ice, liquid water, air) porous media. That is the reason why we specified « freeze-thaw of pore water ».*

L164-167: Since the authors are discussing the speed and efficiency of the OpenFOAM, it would make sense to include further details for the sake of completion, such as the physical dimensions of the domain that was discretized into 1 billion mesh cells and how is this related to the current study?
*The sentence pointed out by the referee does not pretend to discuss the capabilities of OpenFOAM, but only of permaFoam, the permafrost dynamics solver developed within the framework of OpenFOAM. The assessment of the numerical capabilities of permaFoam is the main subject of a previously published paper, 'Permafrost modelling with OpenFOAM®: New advancements of the permaFoam solver' (Orgogozo et al., 2023). The test cases used for assessing the numerical performance of permaFoam in this previous numerical study are not directly related to the study site of the present manuscript, and this has been specified in the updated version (l 179-180).*

- Where else permaFORM is used in the Arctic, more references
*The works using permaFoam are all listed in the bibliography: Orgogozo et al., 2019, Orgogozo et al., 2023. The study site of the present work is the main Arctic site for which permaFoam simulations have been published (Orgogozo et al., 2019, Orgogozo et al., 2023). Simulations with permaFoam have also been performed for the Syrdakh watershed in Eastern Siberia (Orgogozo et al., 2023).*

L128: I would suggest splitting this subsection into two sections, adding, for instance, Model domain. Also, it would help to add a schematic of the model domain with boundary conditions, etc. For the 2D domain, adding a transect can help better understand what exactly the authors are simulating.
*The proposed improvement has been applied (l 184). A detailed description of the simulation domains is available in Orgogozo et al.. (2019).*

L212: Fig 2. "Annual air temperature" should be "Mean annual air temperature" and "Precipitation" should be "Mean annual precipitation" for clarity.

*The proposed improvement has been applied (l 237-238).*

L260: provide some metrics on the plot or in the text as well. Also, please explain why the model failed to capture the dynamics/fluctuation in the observed data.
*Metrics has been added to the text of the updated version, and the related discussion in Supplementary Material A has been more explicitly referred to (l 313-316).*

L275-276: please explain what does "observed increases in air temperature (per 100)" mean here. If this refers to CMIP6 data, that is not observed data.
*In the sentence (L274-276) «These rates of increase […] are lower than the observed increases in air temperature (+1.9°C/100yr for SSP1-2.6 and +7.8°C/100 yr for SSP5-8.5)», the second part refers to the mean annual air temperature increase available in CMIP6 data. Then the word 'observed' will be removed and replaced by 'projected' (l 329).*

L290: Without more details about the topography and model domain, this section does not help in understanding the complexity. The authors used 525K cells, what is the resolution? And why was such a high resolution needed, could the results change with 525K/2 cells – twice coarser resolution?
*The resolution is of 1.2 m laterally, and between 0.25 cm (top) and 16.5 cm (bottom) vertically, since we use a vertically graded mesh in order to save computation time. These information has been added to the Supplementary material B (l 148-151). A convergence study has been made to estimate the resolutions of the spatial discretization required for these calculation. This study is mentioned L300 and is detailed in Supplementary Material B. 525K/4, 525K and 525Kx4 cells meshes have been evaluated. The conclusion of this convergence study is that the 525K cells mesh is needed to obtained results accurate enough for discussing climate change induced variations of Active Layer Thickness. This convergence study have been introduced earlier in the updated version, as part of section 2 (l 198-200).*

L390: How long did one scenario take – real clock time?
*One scenario took between 20 days and 30 days to be completed.*

L315: 2.5 km wide and 10 m thick… are these measurements of the 2D model domain that was divided into 512K cells? This is not that big domain, if I am understanding correctly, this is probably over-discretized. How was this discretization chosen?
*These are the dimensions of the 2D numerical domain, representing one slope (either NAS or SAS) of the Kulingdakan watershed. The discretization is chosen according to the convergence study discussed above, with a convergence criterium based on active layer thickness. Three meshes were used for this convergence study : 1024x128, 2048x256 and 4096x512, labelled as "coarse", "medium", "large" respectively. The differences in computed active layer thickness between medium and large mesh cases were small, with maximum differences of 2.2% for NAS and 1.3% for SAS. These criterium information will be added to the updated supplementary material (l 145-147).*

L370: Please provide relative change/increase for both slopes, which would be more meaningful when comparing the North slope (cold) vs the South slope (warm) permafrost.

*Plots with the relative changes compared to the current value of ALT have been added for both slopes (l 386-389).*

L390: (left figure) In general, the change is less than 2% in all scenarios, however, I would expect the red curve on top of the green curve, as green is a much colder scenario and should be closer to thermal equilibrium as is the case for the SAS (right figure). Please explain.
*This complex behaviour is due to the fact that in its current state the study of thermal equilibrium in 2100 is based on further simulations based on the repetition of the 2100 year, thus encompassing inter-annual variability (e.g.: year 2100 significantly hotter and dryer than previous ones in scenario SSP1-2.6). We updated the thermal equilibrium study by considering 5 years average of the period 2096-2100 as final climatic conditions rather than the year 2100 solely (l 407-409).*

L419: Figure 10. The water would only exist in the active layer which is shallower than the 2 m. It would be more meaningful to plot a time series of the water content in the active layer only. This water content would also depend on the evapotranspiration, and Figure 12 shows ET does not vary significantly in both cases.
*Quantifying the liquid water available on the active layer thickness could lead to interpretation difficulties, since the active layer thickness changes over time. This motivated the choice to averaged liquid water and ice volumetric content over a constant depth superior to the maximum active layer thickness obtained in all the scenarios/slopes. In this way the temporal evolutions of averaged water/ice contents are defined alike and are thus comparable between the different climate scenarios. Figure 12 shows that actual evapotranspiration experiences an increase between current conditions and the 2100 state, quantified between +5 % and +29 % depending on the scenario and slope (Table S3-S4, Supplementary Material C).*

**SELF-MOTIVATED CORRECTION**

An error of parameterization has been identified for the simulations presented in our manuscript submitted to TC. A clipping operation that limited the amplitude of the simulated soil temperature variations between the minimum and maximum monthly values of top soil temperature under current climate was erroneously implemented. So we reran the whole set of simulations with a corrected parameterization, by removing the wrong clipping operation. The comparison between the ill projections (with amplitude limitation) and the correct ones (without amplitude limitation) are presented below.

Using a correct parameterization, i.e. without imposed limitation of the amplitude of simulated soil temperature variations, we obtained detectable but limited changes in the simulation results compared to those obtained with the ill-parameterized simulations included in the currently submitted manuscript. For instance, in the case of the scenario SSP5-8.5, the simulations with correct parameterization leads to an increase of 9.9 % of active layer thickness in NAS and +14.1 % of active layer thickness in SAS in 2100, compared to the simulations with ill parameterization. Meanwhile, no changes in the trends we discussed in our manuscript occur. We replaced all the wrong results and figures with the corrected ones in the updated manuscript. Once again, these modifications do not change the conclusions of our study.

We did not highlight the changes related to this parameterization error in the updated manuscript, in order not to make the localization of changes related to reviewers comments too difficult. Since it involved slight changes in every plots and quantitative information through the manuscript, it would have flooded the changes related to reviewers remarks.

**Comparison between ill-results and correct results:**

*Figure 6: Mean annual temperature evolution at 10cm, 1m, 5m and 10m under the surface for each scenario and slope considered.*

*ILL:*                                                                                          *CORRECT:*

[Figure]

- Left: Soil temperature evolutions with ill parameterization (with soil temperature amplitude limitation to the amplitude of monthly mean top soil temperature under current climate);
- Right: Soil temperature evolutions with correct parameterization (without soil temperature amplitude limitation).

*Figure 7: Annual mean temperature profiles in 2100 and after 30 years of additional cycling of the climatic forcing of this last year.*

ILL :

[Figure]

CORRECT :

[Figure]

*Figure 8: Active layer thickness temporal evolution on North (left) and South (right) aspect Slope of the Kulingdakan watershed obtained from permaFoam simulations under different SSP scenarios.*

ILL:

[Figure]

- Above: Active layer thickness evolutions with ill parameterization (with soil temperature amplitude limitation to the amplitude of monthly mean top soil temperature under current climate);

CORRECT:

[Figure]

- Above: Active layer thickness evolutions with correct parameterization (without soil temperature amplitude limitation).

*Figure 9: Relative change in active layer thickness compared with the year 2100 over 30 years of spin-up of the 2100 climatic conditions.*

ILL :

[Figure]

CORRECT :

[Figure]

*Figure 10: Annual mean of total water content [m 3 of water / m 3 of soil], liquid water content and ice content averaged over 2m depth in different climate projections.*

[Figure]

- Left: Water and ice contents evolutions with ill parameterization (with soil temperature amplitude limitation to the amplitude of monthly mean top soil temperature under current climate);
- Right: Water and ice contents evolutions with correct parameterization (without soil temperature amplitude limitation).

*Figure 11: 2m-depth profiles of annual mean of total water content [m 3 of water / m 3 of soil] in 2100: projections compared to current state.*

*ILL:*

[Figure]

- Above: 2m-depth profiles of annual mean of total water content [m$^3$ of water / m$^3$ of soil] in 2100: projections compared to current state – ill parameterization (with soil temperature amplitude limitation to the amplitude of monthly mean top soil temperature under current climate);

*CORRECT*:

[Figure]

Above: 2m-depth profiles of annual mean of total water content [m$^3$ of water / m$^3$ of soil] in 2100: projections compared to current state – correct parameterization (without soil temperature amplitude limitation).

*Figure 12: Precipitation and actual evapotranspiration evolution over the century*

*ILL:*                        *CORRECT:*

[Figure]

- Left: Actual Evapotranspiration evolutions with ill parameterization (with soil temperature amplitude limitation to the amplitude of monthly mean top soil temperature under current climate);
- Right: Actual Evapotranspiration evolutions with correct parameterization (without soil temperature amplitude limitation).

*Figure 13: (a) Equivalence between simulated active layer thickening by 2100 under climate change (SAS and NAS average) and southward latitudinal shift in current climatic conditions (1997-2019). – latitudinal trend (black line - average over a 1°lat. × 1°long. polygon) and envelops (in grey - min/max over year within the same polygon) extracted from Permafrost_cci (Obu et al., 2021). (b) Representation of the latitudinal southward shift equivalent to each climate scenario's active layer thickening on the regional map.*

ILL:

[Figure]

- Above: Equivalent ALT-Latitude shift based on Permafrost_cci ALT data (Obu et al.,2021) - ill parameterization (with soil temperature amplitude limitation to the amplitude of monthly mean top soil temperature under current climate);

CORRECT:

[Figure]

- Above: Equivalent ALT-Latitude shift based on Permafrost_cci ALT data (Obu et al.,2021) - correct parameterization (without soil temperature amplitude limitation).

| Latidudinal shift compared to Kulingdakan Latitude (64.31°N) | ILL parametrization | CORRECT parametrization |
|---|---|---|
| SSP1-2.6 | -0.46° / 51km | -0.45° / 50km |
| SSP2-4.5 | -1.06° / 118km | -1.17° / 130km |
| SSP3-7.0 | -3.10° / 345km | -5.20° / 578km |
| SSP5-8.5 | -5.06° / 563km | -5.64° / 628km |

*Supplementary material C: tables compiling the main variables change between present conditions and 2100 for the four climate scenarios considered in this paper (SSP1-2.6, SSP2-4.5, SSP3-7.0, SSP5-8.5) for North Aspected Slope (Table C1) and South Aspected Slope (Table C2)*

| *ILL:*
**Variables (NAS)** | **Annual value in present climate** | **Change from present values in projections to 2100** | | | |
| --- | --- | --- | --- | --- | --- |
| | | **SSP1-2.6** | **SSP2-4.5** | **SSP3-7.0** | **SSP5-8.5** |
| Air temperature | -8.2°C | +1.6°C | +3.0°C | +5.6°C | +6.9°C |
| Yearly precipitations | 408mm | +56mm / +14% | +49mm / +12% | +111mm / +27% | +115mm / +28% |
| Maximum snow water equivalent | 108mm | +7mm / +6% | +13mm / +12% | +27mm / +25% | +29mm / +27% |
| Snow season extent | 202days | -6days | -8days | -14days | -17days |
| Soil surface temperature | -3.3°C | +1.4°C | +2.3°C | +4.3°C | +5.2°C |
| Soil temperature (10cm depth) | -4.1°C | +0.9°C | +1.4°C | +2.9°C | +3.4°C |
| Soil temperature (1m depth) | -5.12°C | +0.6°C | +1.0°C | +2.2°C | +2.6°C |
| Soil temperature (5m depth) | -5.06°C | +0.6°C | +1.0°C | +2.2°C | +2.5°C |
| Soil temperature (10m depth) | -4.9°C | +0.6°C | +1.0°C | +2.0°C | +2.5°C |
| Active layer thickness | 64cm | +7.8cm +12% | +11.9cm +19% | +23.9cm +37% | +28.2cm +44% |
| Total water content (averaged over root layer) | 0.510 | $+1.7 \times 10^{-4}$ +0.0% | $-1.2 \times 10^{-2}$ -2.3% | $-1.7 \times 10^{-2}$ -3.3% | $-2.4 \times 10^{-2}$ -4.7% |
| Liquid water content (averaged over root layer) | 0.197 | $+1.2 \times 10^{-2}$ +6.3% | $+1.4 \times 10^{-2}$ +7% | $+2.8 \times 10^{-2}$ +14.1% | $+3.3 \times 10^{-2}$ +16.5% |
| Ice water content (averaged over root layer) | 0.313 | $-1.2 \times 10^{-3}$ -3.9% | $-2.6 \times 10^{-2}$ -8.1% | $-4.5 \times 10^{-2}$ -14.3% | $-5.6 \times 10^{-2}$ -18% |
| Total water content (averaged over 0-2m) | 0.365 | $+3.2 \times 10^{-3}$ +0.9% | $+3.0 \times 10^{-3}$ +0.8% | $+6.7 \times 10^{-3}$ +1.8% | $+6.3 \times 10^{-3}$ +1.7% |
| Liquid water content (averaged over 0-2m) | 0.074 | $+1.1 \times 10^{-2}$ +14.7% | $+1.8 \times 10^{-2}$ +23.8% | $+3.6 \times 10^{-2}$ +49.3% | $+4.4 \times 10^{-2}$ +60.1% |
| Ice water content | 0.291 | $-7.7 \times 10^{-3}$ | $-1.5 \times 10^{-2}$ | $-3.0 \times 10^{-2}$ | $-3.8 \times 10^{-2}$ |

| (averaged over 0-2m) | | -2.7% | -5.0% | -10.2% | -13.1% |
|---|---|---|---|---|---|
| Actual evapotranspiration | 350mm | +40mm / +11% | +52mm / +15% | +98mm /+28% | +100mm / +29% |

*Table C1: main variables changes simulated for the four climate projections in 2100 for north aspected slope - ill parameterization (with soil temperature amplitude limitation to the amplitude of monthly mean top soil temperature under current climate);*

| *CORRECT:* Variables (NAS) | Annual value in present climate | Change from present values in projections to 2100 | | | |
| --- | --- | --- | --- | --- | --- |
| | | SSP1-2.6 | SSP2-4.5 | SSP3-7.0 | SSP5-8.5 |
| Air temperature | -8.2°C | +1.6°C | +3.0°C | +5.6°C | +6.9°C |
| Yearly precipitations | 408mm | +56mm / +14% | +49mm / +12% | +111mm / +27% | +115mm / +28% |
| Maximum snow water equivalent | 108mm | +7mm / +6% | +13mm / +12% | +27mm / +25% | +29mm / +27% |
| Snow season extent | 202days | -6days | -8days | -14days | -17days |
| Soil surface temperature | -3.3°C | +1.4°C | +2.3°C | +4.3°C | +5.2°C |
| Soil temperature (10cm depth) | -4.6°C | +1.2°C | +1.9°C | +3.7°C | +4.4°C |
| Soil temperature (1m depth) | -5.6°C | +1.0°C | +1.5°C | +2.9°C | +3.4°C |
| Soil temperature (5m depth) | -5.6°C | +1.0°C | +1.5°C | +2.8°C | +3.2°C |
| Soil temperature (10m depth) | -5.5°C | +0.9°C | +1.5°C | +2.7°C | +3.2°C |
| Active layer thickness | 63cm | +8.8cm +14% | +14.5cm +23% | +30.9cm +49% | +38.5cm +61% |
| Total water content (averaged over root layer) | 0.510 | $1.1 \times 10^{-4}$ +0.0% | $-1.2 \times 10^{-2}$ -2.3% | $-1.7 \times 10^{-2}$ -3.4% | $-2.4 \times 10^{-2}$ -4.7% |
| Liquid water content (averaged over root layer) | 0.198 | $1.2 \times 10^{-2}$ +5.9% | $1.3 \times 10^{-2}$ +6.5% | $2.7 \times 10^{-2}$ +13.8% | $3.2 \times 10^{-2}$ +16.3% |
| Ice water content (averaged over root layer) | 0.312 | $-1.2 \times 10^{-2}$ -3.7% | $-2.5 \times 10^{-2}$ -7.9% | $-4.4 \times 10^{-2}$ -14.2% | $-5.6 \times 10^{-2}$ -18.0% |
| Total water content (averaged over 0-2m) | 0.364 | $+3.5 \times 10^{-3}$ +1.0% | $+3.9 \times 10^{-3}$ +1.1% | $+9.4 \times 10^{-3}$ +2.6% | $+9.3 \times 10^{-3}$ +2.6% |
| Liquid water content (averaged over 0-2m) | 0.072 | $+1.2 \times 10^{-2}$ +17.3% | $+2.0 \times 10^{-2}$ +28.4% | $+4.5 \times 10^{-2}$ +62.4% | $+5.6 \times 10^{-2}$ +77.8% |

| | | | | | |
|---|---|---|---|---|---|
| Ice water content (averaged over 0-2m) | 0.292 | -8.9x10-03 -3.1% | -1.7x10$^{-2}$ -5.7% | -3.6x10$^{-2}$ -12.2% | -4.7x10$^{-2}$ -16.0% |
| Actual evapotranspiration | 351mm | +35mm/+10% | +51mm/+14% | +108mm/ +31% | +123mm/+35% |

*Table C1: main variables changes simulated for the four climate projections in 2100 for north aspected slope - correct parameterization (without soil temperature amplitude limitation).*

| *ILL:*
**Variables (SAS)** | **Annual value in present climate** | **Change from present values in projections to 2100** | | | |
| --- | --- | --- | --- | --- | --- |
| | | **SSP1-2.6** | **SSP2-4.5** | **SSP3-7.0** | **SSP5-8.5** |
| Air temperature | -8.2°C | +1.6°C | +3.0°C | +5.6°C | +6.9°C |
| Yearly precipitations | 408mm | +56mm/+14% | +49mm / +12% | +111mm / +27% | +115mm / +28% |
| Maximum snow water equivalent | 108mm | +7mm / +6% | +13mm / +12% | +27mm / +25% | +29mm / +27% |
| Snow season extent | 202days | -6days | -8days | -14days | -17days |
| Soil surface temperature | -2.6°C | +1.5°C | +2.3°C | +4.4°C | +5.2°C |
| Soil temperature (10cm depth) | -3.1°C | +1.1°C | +1.7°C | +3.4°C | +4.0°C |
| Soil temperature (1m depth) | -4.15°C | +1.0°C | +1.5°C | +2.9°C | +3.3°C |
| Soil temperature (5m depth) | -4.11°C | +0.9°C | +1.5°C | +2.4°C | +2.7°C |
| Soil temperature (10m depth) | -4.0°C | +0.9°C | +1.5°C | +2.3°C | +2.5°C |
| Active layer thickness | 99cm | +13cm +13% | +20.0cm +20% | +36.3cm +37% | +45.2cm +46% |
| Total water content (averaged over root layer) | 0.375 | $-1.6 \times 10^{-2}$ -4.3% | $-2.1 \times 10^{-2}$ -5.6% | $-3.2 \times 10^{-2}$ -8.5% | $-3.7 \times 10^{-2}$ -9.7% |
| Liquid water content (averaged over root layer) | 0.153 | $+1.1 \times 10^{-3}$ +0.7% | $+3.5 \times 10^{-3}$ +2.3% | $+1.2 \times 10^{-2}$ +8.0% | $+1.5 \times 10^{-2}$ +9.8% |
| Ice water content (averaged over root layer) | 0.222 | $-1.7 \times 10^{-2}$ -7.7% | $-2.4 \times 10^{-2}$ -11.0% | $-4.4 \times 10^{-2}$ -19.9% | $-5.1 \times 10^{-2}$ -23.1% |
| Total water content (averaged over 0-2m) | 0.343 | $-1.8 \times 10^{-2}$ -5.4% | $-1.7 \times 10^{-2}$ -5.0% | $-3.4 \times 10^{-2}$ -9.8% | $-3.7 \times 10^{-2}$ -10.8% |
| Liquid water content (averaged over 0-2m) | 0.090 | $+6.6 \times 10^{-3}$ +7.3% | $+1.5 \times 10^{-2}$ +16.2% | $+3.4 \times 10^{-2}$ +37.7% | $+4.6 \times 10^{-2}$ +50.7% |
| Ice water content (averaged over 0-2m) | 0.253 | $-2.5 \times 10^{-2}$ -9.9% | $-3.2 \times 10^{-2}$ -12.6% | $-6.8 \times 10^{-2}$ -26.8% | $-8.3 \times 10^{-2}$ -32.7% |

| Actual evapotranspiration | 364mm | +18mm/+5% | +34mm/+9% | +76mm/+21% | +94mm/+26% |

*Table C2: main variables changes simulated for the four climate projections in 2100 for south aspected slope - ill parameterization (with soil temperature amplitude limitation to the amplitude of monthly mean top soil temperature under current climate);*

| CORRECT: Variables (SAS) | Annual value in present climate | Change from present values in projections to 2100 | | | |
|---|---|---|---|---|---|
| | | SSP1-2.6 | SSP2-4.5 | SSP3-7.0 | SSP5-8.5 |
| Air temperature | -8.2°C | +1.6°C | +3.0°C | +5.6°C | +6.9°C |
| Yearly precipitations | 408mm | +56mm/+14% | +49mm / +12% | +111mm / +27% | +115mm / +28% |
| Maximum snow water equivalent | 108mm | +7mm / +6% | +13mm / +12% | +27mm / +25% | +29mm / +27% |
| Snow season extent | 202days | -6days | -8days | -14days | -17days |
| Soil surface temperature | -2.6°C | +1.5°C | +2.3°C | +4.4°C | +5.2°C |
| Soil temperature (10cm depth) | -3.3°C | +1.4°C | +2.1°C | +4.2°C | +5.0°C |
| Soil temperature (1m depth) | -4.4°C | +1.3°C | +1.8°C | +3.5°C | +4.0°C |
| Soil temperature (5m depth) | -4.4°C | +1.2°C | +1.8°C | +2.9°C | +3.1°C |
| Soil temperature (10m depth) | -4.3°C | +1.2°C | +1.8°C | +2.7°C | +2.9°C |
| Active layer thickness | 100cm | +12.5cm +13% | +22.6cm +23% | +46.5cm +47% | +65.1cm +65% |
| Total water content (averaged over root layer) | 0.375 | $-1.8\times10^{-2}$ -4.9% | $-2.2\times10^{-2}$ -6.0% | $-3.2\times10^{-2}$ -8.6% | $-3.5\times10^{-2}$ -9.4% |
| Liquid water content (averaged over root layer) | 0.152 | $+7.6\times10^{-4}$ +0.5% | $+3.9\times10^{-3}$ +2.5% | $+1.2\times10^{-2}$ +8.2% | $+1.6\times10^{-2}$ +10.3% |
| Ice water content (averaged over root layer) | 0.223 | $-1.9\times10^{-2}$ -8.5% | $-2.6\times10^{-2}$ -11.8% | $-4.5\times10^{-2}$ -20.1% | $-5.1\times10^{-2}$ -22.9% |
| Total water content (averaged over 0-2m) | 0.339 | $-1.6\times10^{-2}$ -4.6% | $-1.8\times10^{-2}$ -5.4% | $-3.2\times10^{-2}$ -9.5% | $-3.4\times10^{-2}$ -9.9% |
| Liquid water content (averaged over 0-2m) | 0.089 | $+8.2\times10^{-3}$ +9.2% | $+1.8\times10^{-2}$ +20.4% | $+4.4\times10^{-2}$ +49.8% | $+6.4\times10^{-2}$ +72.4% |
| Ice water content (averaged over 0-2m) | 0.250 | $-2.4\times10^{-2}$ -9.6% | $-3.6\times10^{-2}$ -14.5% | $-7.7\times10^{-2}$ -30.6% | $-9.8\times10^{-2}$ -39.1% |
| Actual | 361mm | +19mm/+5% | +37mm/+10% | +82mm/+23% | +108mm/+30% |

evapotranspiration

*Table C2: main variables changes simulated for the four climate projections in 2100 for south aspected slope - correct parameterization (without soil temperature amplitude limitation).*

---

## Referee Report (RR1)

**Review of the revised manuscript "Future permafrost degradation under climate change in a headwater catchment of Central Siberia: quantitative assessment with a mechanistic modelling approach" by Xavier et al. submitted to The Cryosphere**

In the revised version the authors have addressed the comments raised by both reviewers. The readability was greatly improved and clarifications were added to the text. While I think the manuscript is in a much better shape now, I believe some comments are not addressed adequately and require a little bit more work.

1. L58: this sentence is still confusing. Do you mean something like "and thereby covers 55% of the total global permafrost area"? Please clarify
2. L95: the abbreviation CMIP6 shows up in the main text here for the first time, but you only define it for the first time in L213, please adjust.
3. Fig. 10: Rev #2 had a good suggestion to better illustrate the water content with varying active layer depths. This is a valid comment, especially considering the differences in ALT between NAS and SAS. I agree with the authors that it is not meaningful to use maximum ALT as a reference depth, but an additional plot (maybe in the Supplementary Material) showing a time series of the liquid and ice content within the thaw depth (depth until T < 0°C, changes throughout the season) would be interesting. This could be done for the present conditions, a year in the middle of the century and by the end of the simulation (or in the equilibrium simulations). This would be important to interpret the availability of liquid water throughout the season to better align it with the growing season.
4. Fig. 10: The figure caption is not very informative as it is. Maybe change it to something like "Annual mean of total water content [m 3 of water / m3 of soil] partitioned into liquid (blue) and ice (grey) content...."
5. L455-458: The explanation of the processes driving moisture distribution is still insufficient in my opinion. They are not explained in the subsequent text. Rather, a direction of water movement is given but without explaining what is causing it. This still needs work.
6. L503-506: The discussion on the water flux changes is insufficient given the description of the results. With a more careful description of this in the result section, the discussion can be improved accordingly.

Supplement:

L185: What is the reference for the geothermal heat flux boundary condition?

L237: The text is copied from the response letter. Please remove "This comparison will be added to the supplementary material."

Generally, I find the referencing to the Supplementary Material hard to follow. With some restructuring, the references to the individual text parts can be improved (e.g., not starting with Supplementary Material B in L130 and more clearly stating which part of it refers to what is being said in the main text).

---

## Author Response (AR2)

Dear referee,

We thank you for these additional comments about the updated version of our manuscript. The modifications for taking into account these comments are included in the attached version, highlighted in green. Please find below our answer to the points you raised.

Best regards,

The authors.

1. L58: this sentence is still confusing. Do you mean something like "and thereby covers 55% of the total global permafrost area"? Please clarify

The sentence is clarified by including the proposed wording.

2. L95: the abbreviation CMIP6 shows up in the main text here for the first time, but you only define it for the first time in L213, please adjust.

The abbreviation is expanded in the introduction and reminded in the section 2.4.

3. Fig. 10: Rev #2 had a good suggestion to better illustrate the water content with varying active layer depths. This is a valid comment, especially considering the differences in ALT between NAS and SAS. I agree with the authors that it is not meaningful to use maximum ALT as a reference depth, but an additional plot (maybe in the Supplementary Material) showing a time series of the liquid and ice content within the thaw depth (depth until T < 0°C, changes throughout the season) would be interesting. This could be done for the present conditions, a year in the middle of the century and by the end of the simulation (or in the equilibrium simulations). This would be important to interpret the availability of liquid water throughout the season to better align it with the growing season.

We do agree that illustrating the liquid water available in the active layer is a key element to provide insight into possible future vegetation evolution.

The additional figure below shows the evolution of the annual mean of the total water content, partitioned into ice and liquid, averaged over the active layer (different for each year, scenario and slope), as proposed by Rev#2. However, the combined change in both the water content and the integration thickness (the ALT) makes the interpretation difficult, thus we did not include this Figure in the submitted material.

In order to quantify clearly the change in liquid water available for vegetation uptake, we plotted the evolution of the integral of the liquid water content over the surficial thawed layer (above the T=0°C isotherm). This proxy of liquid water availability is thus expressed in meter. This plot is made for each scenario and slope, for the years 2014, 2050 and 2100. The resulting figure (Supplementary material, l 250) shows both an increase of the maximum liquid water available during the year (up to +64 % in NAS and up to +61% in SAS, obtained under SSP5-8.5, compared to the present value), and an extension of the period of availability of liquid water during the year (up to +39 days for NAS, and +35 days for SAS, obtained under SSP5-8.5, compared to the present value). A quantitative summary of these results is given in the additional table below. This figure is added to Supplementary Material D – Seasonal change in liquid water available for vegetation uptake, and referred to in the results section (l 446 -447) and in the discussion section (l 508-510).

[Figure]

Additional figure: Annual mean of total water content [m3 of water / m3 of soil] partitioned into liquid (blue) and ice (grey) water content averaged over the active layer in different climate projections.

| Variables | Annual value in present climate | Change from present values in projections to 2100 | | | |
| --- | --- | --- | --- | --- | --- |
| | | SSP1-2.6 | SSP2-4.5 | SSP3-7.0 | SSP5-8.5 |
| Maximum liquid water content available (NAS) | 19.6 cm | +3.0 cm +15 % | +4.7cm +24 % | +10.1cm +52 % | +12.6cm +64 % |
| Maximum liquid water content available (SAS) | 27.5 cm | +2.5 cm +9 % | +4.5cm +17 % | +10.7cm +39 % | +16.7cm +61 % |
| Days with more than 1cm of liquid water available for vegetation uptake (NAS) | 140 days | +10 days +7 % | +18 days +13 % | +31 days +22 % | +39 days +28 % |
| Days with more than 1cm of liquid water available for vegetation uptake (SAS) | 152 days | + 7 days +5 % | +14 days +9 % | +28 days +18 % | +35 days +23 % |

Additional table: quantitative summary of changes in liquid water availability between current conditions and the four climate projections for 2100 used in this study.

4. Fig. 10: The figure caption is not very informative as it is. Maybe change it to something like "Annual mean of total water content [m 3 of water / m3 of soil] partitioned into liquid (blue) and ice (grey) content...."

The caption is modified using the proposed wording.

5. L455-458: The explanation of the processes driving moisture distribution is still insufficient in my opinion. They are not explained in the subsequent text. Rather, a direction of water movement is given but without explaining what is causing it. This still needs work.

The presentation of Figure 11 has been rewritten (l 459-468) in order to put forward a hypothesis regarding the processes driving moisture distribution in the slopes of Kulingdakan watershed, with a focus on the role of the contrast of rooting depth between SAS and NAS. Rooting depth has been previously identified as a key control on the thermo-hydrological regime in the active layers of the study site (Orgogozo et al., 2019), and thus proposing an interpretation of the water fluxes based on it seems reasonable to us.

6. L503-506: The discussion on the water flux changes is insufficient given the description of the results. With a more careful description of this in the result section, the discussion can be improved accordingly.

The discussion of the water fluxes changes has been slightly extended (l 511-512 and l 516-520), putting forward the impact of the drying of the root layers.

Supplement:
L185: What is the reference for the geothermal heat flux boundary condition?

The reference is added in the supplementary material as well (Duchkov et al., 1997).

L237: The text is copied from the response letter. Please remove "This comparison will be added to the supplementary material."

This residual from the response letter is removed.

Generally, I find the referencing to the Supplementary Material hard to follow. With some restructuring, the references to the individual text parts can be improved (e.g., not starting with

Supplementary Material B in L130 and more clearly stating which part of it refers to what is being said in the main text).

The names of the parts are included in the body of the text when references are made to the supplementary materials.

---

## Author Response (AR3)

Dear reviewer and editorial board,

Thank you for reviewing our paper submitted for publication in The Cryosphere, and for taking into account the exchanges in the previous stages of this process.

The syntax and typographical errors you pointed out have been corrected. The suggested wording for line 459 is much clearer and has been used as is.

We are pleased to be able to make this contribution to The Cryosphere. We wish you all the best for the future.

The authors.